# Local Time Dependence of Polar Mesospheric Clouds: A model study

Francie Schmidt[1], Gerd Baumgarten[1], Uwe Berger[1], Jens Fiedler[1], and Franz-Josef Lübken[1]

[1]Leibniz-Institute of Atmospheric Physics, Rostock University, Kühlungsborn, Germany

*Correspondence to:* Gerd Baumgarten (baumgarten@iap-kborn.de)

**Abstract.**

The Mesospheric Ice Microphysics And tranSport model (MIMAS) is used to study local time (LT) variations of polar mesospheric clouds (PMC) in the northern hemisphere during the period from 1979 to 2013. We investigate the tidal behavior of brightness, altitude and occurrence frequency and find a good agreement between model and lidar observations. At the peak of the PMC layer the mean ice radius varies from 35 to 45 nm and the mean number density from 80 to 150 cm$^{-3}$ throughout the day. We also analyze PMC in terms of ice water content (IWC) and show that only amplitudes of local time variations in IWC are sensitive to threshold conditions whereas phases are conserved. In particular, relative local time variations decrease with larger thresholds. Local time variations also depend on latitude. In particular, absolute local time variations increase towards the pole. Furthermore, a phase shift exists towards the pole which is independent of the threshold value. In particular, the IWC maximum moves backward in time from 8 LT at mid latitudes to 2 LT at high latitudes. The persistent features of strong local time modulations in ice parameters are caused by local time structures in background temperature and water vapor. For a single year local time variations of temperature at 69°N are in a range of 6 K near 83 km altitude. At sublimation altitudes the water vapor variation is about 7 ppmv, leading to a change of the saturation ratio by a factor of about 2 throughout the day.

## 1 Introduction

Polar mesospheric clouds (PMC), also known as noctilucent clouds (NLC), consist of water-ice crystals. They occur at mid to high latitudes around 83 km altitude (e.g. Jesse, 1896; Gadsden and Schröder, 1998; Lübken et al., 2008). Such clouds form in summer in a supersaturated cold atmosphere with temperatures below 150 K and are sensitive to water vapor and mesospheric temperatures. Therefore, PMC are thought to be sensitive indicators of climate change in the middle atmosphere (e.g. Thomas, 1996; Berger and Lübken, 2015; Hervig et al., 2016a). PMC often show a rich variability which provides information about thermal and dynamical processes on thermal background fields (Witt, 1962). The clouds have been shown to be subject to persistent local time variations (e.g. von Zahn et al., 1998; Chu et al., 2003; Fiedler et al., 2005). These variations were attributed to atmospheric thermal tides. Such tidal oscillations are globally forced due to absorption of solar irradiance throughout the day. While semidiurnal tides are dominantly generated through absorption of solar ultraviolet radiation by stratospheric ozone, water vapor in the troposphere absorbs solar radiation in the near-infrared bands forcing mainly the diurnal tidal component

(Lindzen and Chapman, 1969). Generally, these tidal waves propagate upwards with exponential growth in amplitude, and are therefore also present at PMC altitudes in the summerly mesopause region at high latitudes.

A variety of spaceborne experiments have observed PMC since the late 20th century (e.g. Stevens et al., 2010; Russell et al., 2014; Hervig and Stevens, 2014). Many of these experiments are on satellites with sun-synchronous orbits and therefore only allow observations at fixed local times. The Solar Backscattered Ultraviolet Instruments (SBUV) on-board the National Oceanic and Atmospheric Administration (NOAA) satellites provide a data set of more than 35 years of PMC observations (e.g. Thomas et al., 1991). This data set was recorded by eight separate instruments with changing viewing conditions and different local times which introduces uncertainties in the long-term analysis when creating a single data set. Also the Solar Occultation For Ice Experiment (SOFIE) and the Cloud Imaging and Particle Size (CIPS) instrument on-board the Aeronomy of Ice in the Mesosphere (AIM) satellite perform observations in a sun-synchronous orbit. The Ozone Monitoring Instrument (OMI) on-board the Aura satellite is able to measure PMC at different local times, but only part of the diurnal cycle is covered, i.e. the afternoon is missing (DeLand et al., 2011). In order to quantify long-term natural or anthropogenic changes in PMC, it is therefore essential to understand their variations over the diurnal cycle (DeLand and Thomas, 2015).

In contrast to satellite measurements, ground-based measurements are geographically restricted but have the ability to cover a full local time cycle. E.g. variations of PMC occurrence frequency and brightness as function of local time have been observed in detail with lidar instruments (von Zahn et al., 1998; Chu et al., 2006; Fiedler et al., 2005, 2009, 2011, 2017; Gerding et al., 2013). All these data show evidence of a large PMC brightness variability with local time.

In this paper we discuss results from a three-dimensional Lagrangian transport model for PMC called MIMAS (Mesospheric Ice Microphysics And tranSport model), see also the data description in Berger and Lübken (2015). MIMAS covers the latitude and altitude range of PMC and the entire PMC season with a high temporal resolution. This allows for example to calculate latitude-dependent local time adjustments to retrieve PMC parameters with the observational filter of satellite instruments. In the next section we describe some important aspects of the MIMAS model which are relevant for the simulation of seasonal and local time variations in PMC. Here, we also describe some mean atmospheric background conditions and we characterize local time variations of background temperature and water vapor as calculated by the model. Furthermore we give an overview of local time variations in backscatter (section 3), ice water content (section 4), and ice particle radius, number density and ice mass density (section 5) seen in MIMAS and compared to lidar and satellite observations. Finally, we discuss the latitudinal dependencies ( section 6) of local time variations in IWC and their possible implications when analyzing satellite data at fixed local times.

## 2 The MIMAS ice model

### 2.1 Model description

The MIMAS model is a 3-dimensional Lagrangian transport model designed specifically to model ice particles in the meso-sphere/lower thermosphere (MLT) region. MIMAS is limited from mid to high latitudes ($45-90°$N) with a horizontal grid of $1°$ in latitude and $3°$ in longitude, and a vertical resolution of $100\,m$ from 77.8 to 94.1 km (163 levels).

Typically, MIMAS calculates a complete PMC season from mid of May to end of August. Each of the seasonal simulations starts with the same water vapor distributions on constant pressure levels (Berger and Lübken, 2015). Then, the background water vapor is transported by 3-d winds, mixed by turbulent diffusion, and reduced by photo-dissociation from solar ultraviolet radiation. We use Lyman-$\alpha$ as a proxy for solar activity (available at http://lasp.colorado.edu/lisird/lya/).

Simultaneously, 40 million condensation nuclei (dust particles) are transported according to 3–d background winds, particle eddy diffusion, and sedimentation. The radii of the dust particles in the model vary according to a Hunten distribution between 1.2 and 3.6 nm (Berger and von Zahn, 2007). While each of the 40 million particles is transported on an individual 3–d trajectory with a time step of 45 s, a single dust particle will nucleate or an already existing ice particle will further grow, respectively, whenever the temperature and water vapor concentration of the background atmosphere provide conditions of supersaturation. In the case of undersaturated conditions a preexisting ice particle will start to sublimate. The local formation, growth, and sublimation of all ice particles are interactively coupled to the local background water vapor concentration which leads to a redistribution of $H_2O$ with local freeze drying and water supply (von Zahn and Berger, 2003; Kiliani, 2014; Berger and Lübken, 2015).

In MIMAS temperatures, densities, pressure and wind fields are prescribed using hourly output data from the Leibniz Institute Middle Atmosphere (LIMA) model which especially aims to represent the thermal structure around mesopause altitudes (Berger, 2008). LIMA is a fully nonlinear, global, and three-dimensional Eulerian grid point model taking into account major processes of radiation, chemistry, and transport. LIMA extends from the ground to the lower thermosphere (0–150 km), and applies a triangular horizontal grid structure with 41804 grid points in every horizontal layer ($\Delta x \approx \Delta y \approx 110$ km). This allows to resolve the fraction of the large-scale internal gravity waves with horizontal wavelengths of $\geq 500$ km.

LIMA is nudged to tropospheric and stratospheric reanalysis data available from the European Centre for Medium-Range Weather Forecasts (ECMWF), Reading, United Kingdom. LIMA incorporates the 40 year ECMWF reanalysis data set (ERA-40) from 1960 to 2002 and ECMWF operational analysis thereafter. The nudging coefficient is altitude dependent with a constant value of $1/3.5$ days$^{-1}$ from the ground to the middle stratosphere (35 km). Above 35 km, the coefficient linearly decreases to zero until 45 km. The nudging of ECMWF data introduces short-term and year-to-year variability. Above approximately 40 km, carbon dioxide and ozone concentrations as well as solar activity vary with time. For $CO_2$ we have used a monthly mean time series for the entire period (1961–2013) as measured at Mauna Loa (from http://www.esrl.noaa.gov/gmd/ccgg/trends). For ozone, we take a temporal variation in the height region of the upper stratosphere and lower mesosphere (40–65 km) into account. More precisely, we have used relative anomalies at 0.5 hPa from 1979 to 2013 as measured by SBUV satellite instruments (from http://acd-ext.gsfc.nasa.gov/Data-services/merged/), for more details see Lübken et al. (2013). Before 1979 ozone data are taken from the World Meteorological Organization (WMO) report. Finally, daily Lyman-$\alpha$ fluxes from January 1961 until December 2013 are taken as a proxy for solar activity.

## 2.2 Mean state and local time variations of atmospheric background temperature and water vapor

Certainly, background conditions of temperature and background water vapor are of overriding importance controlling ice formation in the mesopause region. In the following we will shortly summarize some main MIMAS results of mean state and

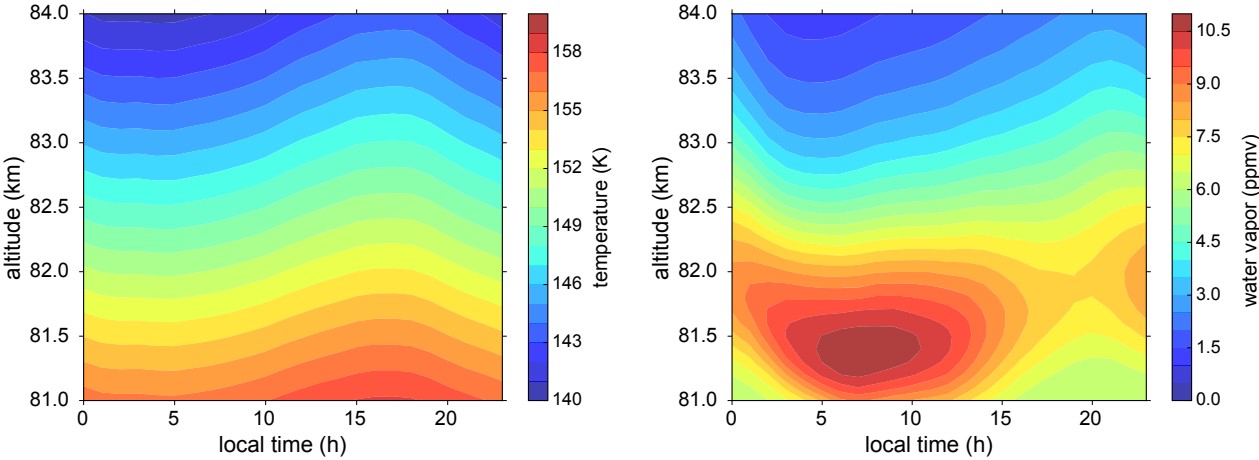

**Figure 1.** Local time variation derived from monthly and zonal means of temperature (left) and water vapor (right) in the latitude band 67°–71°N for July 2009, see text for more details.

local time variation of the background. We show in Figure 1 examples of monthly and zonally averaged temperature and water vapor fields as a function of local time in the northern hemisphere for July 2009. We choose the altitude region 81–84 km in order to resolve typical background conditions of temperature and water vapor concentrations at PMC heights. We selected a single year, namely 2009, to be unaffected by possible long-term variations of the local time behavior. In addition, the year
2009 was analyzed in detail by previous studies (Kiliani et al., 2013; Kiliani et al., 2015). The monthly average shown in Figure 1 has been determined using a one hourly output of temperatures and water vapor from the MIMAS $1° \times 3° \times 100$ m latitude-longitude-height grid. For each hourly data set, the actual longitudinal position on a latitudinal circle is transformed to a uniform local time. Hence, our local time resolution is defined by the number of 120 longitudinal grid points for a given hourly data set. Finally, we calculate the monthly July average from 31 (days) times 24 samples per day. We note that this averaging
process resolves the mean sun-synchronous part of migrating tidal oscillations. In the following we name this procedure as 'method 1' that allows to identify mean local time variations basing on a monthly zonal average.

Another possibility to examine local time structures is to analyze straightforward time series of a single day based on hourly data for individual latitudinal and longitudinal grid points ('method 2'). We then estimate from each daily data sample specific parameters of mean and maximum/minimum values including corresponding times. Additionally, sinusoidal fits are applied
to this daily sample in order to calculate 24 h, 12 h and 8 h tidal amplitudes and phases. This procedure is repeated for every grid point taking into account the difference in local time on various longitudinal positions, and for every day during July. After averaging, we finally get mean values of parameters that describe monthly local time variations on the basis of local daily time series. Generally, method 1 generates smaller estimates of mean local time variations than method 2 since local time parameters are determined from a highly smoothed state in method 1. Conversely, method 2 uses single day time series,
and therefor also records day-to-day variations of daily fluctuations which depends not only on variable tidal wave activity but

| Latitude | Height | Mean | Max | Min | LT(Max) | LT(Min) | $A_{24}$ | $A_{12}$ | $P_{24}$ | $P_{12}$ |
|---|---|---|---|---|---|---|---|---|---|---|
| 54°N | 83 | 149 | 154 | 144 | 15.0 | 6.6 | 3.9 | 1.9 | 17.4 | 1.4 |
| 69°N | 83 | 147 | 150 | 144 | 17.0 | 4.4 | 1.9 | 1.4 | 16.8 | 5.6 |
| 78°N | 83 | 147 | 149 | 145 | 15.8 | 2.0 | 1.2 | 0.8 | 15.2 | 6.0 |
| 54°N | 90 | 163 | 178 | 148 | 20.4 | 14.6 | 5.5 | 6.2 | 0.0 | 8.0 |
| 69°N | 90 | 135 | 148 | 126 | 13.2 | 23.6 | 5.0 | 5.7 | 13.2 | 2.2 |
| 78°N | 90 | 126 | 133 | 121 | 15.2 | 1.4 | 2.9 | 3.2 | 14.8 | 4.2 |

**Table 1.** Local time variation derived from daily data of temperature [K] for two heights [km] at different latitudes for July 2009, see text for more details. Mean: mean temperature over a daily cycle ; Max: maximum temperature over a daily cycle ; Min: minimum temperature over a daily cycle; LT(Max): local time (LT) in hours of Max; LT(Min): local time (LT) in hours of Min; $A_{24}$: diurnal amplitude from a harmonic fit including 24h and 12h components; $A_{12}$: same but for the semidiurnal amplitude; $P_{24}$: diurnal phase of $A_{24}$ in LT hours of maximum; $P_{12}$: same but for semidiurnal phase.

also on variable planetary and large scale gravity wave activity, e.g. as observed by Baumgarten et al. (2018). However, our MIMAS simulations are driven by hourly inputs and not by a monthly zonal mean state. For this reason results from method 2 should better describe mean local time fluctuations of background conditions that effect ice formation. Table 1 summarizes some relevant numbers that describe the mean state and local time fluctuations of temperature resulting from method 2.

We begin with a short discussion of the general mean background state of temperatures. Both averaging procedures from method 1 (Figure 1) and 2 (Table 1, 3rd column) result in identical monthly mean values of temperatures. The modeled temperatures closely match observed mesopause temperatures and altitudes. Monthly mean MIMAS temperatures at 69°N are very similar to the observed temperature climatology derived from rocket (falling spheres) measurements at ALOMAR (69°N) during summer (Lübken, 1999; Schöch et al., 2008). For example, the minimum temperature is $\sim 130\,\mathrm{K}$ in MIMAS compared to
$\sim 130\,\mathrm{K}$ in the climatological observations for July, and also the summer mesopause altitude is basically identical ( $\sim 88$ km). At typical NLC heights at 83 km mean MIMAS temperatures are a bit higher with $\sim 147\,\mathrm{K}$ compared to observed $\sim 145\,\mathrm{K}$. The MIMAS summer mesopause at 78°N (89 km/124 K) is colder and higher compared to lower latitudes. Lidar measurements of temperatures were performed in the upper mesosphere/lower thermosphere (MLT) at Spitsbergen (78°N) in the years 2001– 2003. The July observations show that the summer mesopause is located at 90 km and is as cold as 122 K (Höffner and Lübken,
2007). At lower latitudes at 54°N the MIMAS mesopause is significantly lower (86 km), warmer (144 K), and less pronounced. Again, lidar observations of temperatures confirm these model results with for example a mean July mesopause (86 km/147 K) at Kühlungsborn (54°N) (Gerding et al., 2008). So far, we validated model temperatures of the summer mesopause region only with observational climatologies obtained from groundbased lidar facilities and rocket measurements which we think provide reliable data sets for the high latitude MLT-region. Furthermore, groundbased measurements with meteor radars indicate low
temperatures around 90 km in summer typically in a range of 150–170 K at 54°N and 120–140 K at 69°N (Singer et al., 2003, 2005). Calculated temperatures from MIMAS fit to these observations, see Table 1. Stevens et al. (2017) also published temperatures for July 2009 observed by the SOFIE satellite instrument which show systematic and large differences compared to

lidar data. For example, SOFIE temperatures indicate a mesopause at 88 km similar to lidars but with a mesopause temperature of $\sim 140\,\mathrm{K}$ which is a difference of $\sim 10\,\mathrm{K}$. We note that such a warm mesopause would dramatically prevent ice nucleation and growth in MIMAS with resulting highly underestimated ice masses.

Figure 1 and Table 1 also show mean daily temperature fluctuations. Looking at Figure 1, local time variations calculated
with method 1 have a value about 2–3 K near 83 km at 69°N. Applying our preferred averaging procedure from method 2 yields systematically larger local time variations, see Table 1. The analysis shows that in the height region 83–90 km local time variations of temperature decrease towards the pole, i.e. 10–30 K at 54°N, 6–22 K at 69°N, and 4–12 K at 78°N. Generally, the tidal analysis of temperatures indicates that diurnal and semidiurnal tides are mainly present whereas the terdiurnal component can be neglected. Thermal amplitudes increase with altitude and decrease with poleward direction as has been discussed in
Stevens et al. (2017). Absolute values of diurnal and semidiurnal amplitudes from MIMAS are in the same order as has been calculated in the model study by Stevens et al. (2010, 2017). Also tidal temperature variations derived from Meteor radar observations around 90 km in summer show diurnal (semidiurnal) amplitudes of about 7 K (5 K) at 54°N, and amplitudes of about 4–8 K (2–4 K) at higher latitudes 69°N (Singer et al., 2003). These observations match the size of amplitudes estimated by MIMAS, see Table 1.

At PMC altitudes near 83 km diurnal tidal amplitudes are up to a factor two stronger than semidiurnal amplitudes. This means that local variations of temperatures are mainly affected by diurnal tidal modes. At mesopause altitudes diurnal and semidiurnal amplitudes get larger and are of similar size.

We also compared the phase structures as calculated by the two averaging procedures from method 1 and 2, and find that phases of maximum and minimum values as well as tidal phases remain almost unchanged. Interestingly, temperature phases
change with latitude at PMC altitudes. Particularly, the local time of the daily minimum (Table 1, 7th column) is shifted backwards in time towards higher latitudes from 6.6 LT (54°N) to 4.4 LT (69°N) and 2.0 LT (78°N). Contrary to the shift of the minimum, the time of temperature maximum seems to occur steadily always between 15 LT and 17 LT. The superposition of diurnal and semidiurnal thermal tides causes predominantly lower temperatures during early morning hours and higher temperatures during afternoon hours, respectively.

Beside temperatures, water vapor plays an essential role for PMC formation. Figure 1 shows water vapor mixing ratios from MIMAS ice simulations at latitudes 67°–71°N for July 2009. In addition, Table 2 describes numbers, using method 2, of latitudinal dependencies for daily variations of water vapor. At 69°N the mean vertical water vapor profile maximizes at 81.5 km with 8 ppmv where ice particles sublimate and create a zone of enhanced hydration. SOFIE observations of water vapor at 73°N show a similar vertical structure with a water vapor peak of 8 ppmv near $\sim 83\,\mathrm{km}$ (Hervig et al., 2016b).
From Table 2 we find that effects of hydration (sublimation of ice) at 81.5 km and dehydration (freeze drying) near 84 km are intensified towards higher latitudes since colder mesopause temperatures permit larger nucleation rates of ice particles, and larger sedimentation paths lead to enhanced growth of ice particles that causes enhanced sublimation.

MIMAS results indicate that local time variations of water vapor in terms of absolute values are much stronger than thermal local time variations. At 69°N local time variability of background water vapor can reach values up to 7 ppmv at 81.5 km
which is in the order of a 100 % variation. Consequently, tidal amplitudes of water vapor from harmonic fits show large tidal

| Latitude | Height | Mean | Max | Min | LT(Max) | LT(Min) | $A_{24}$ | $A_{12}$ | $P_{24}$ | $P_{12}$ |
|---|---|---|---|---|---|---|---|---|---|---|
| 54°N | 81.5 | 4.4 | 5.4 | 3.5 | 7.0 | 13.0 | 0.74 | 0.37 | 0.8 | 7.4 |
| 69°N | 81.5 | 8.0 | 12.6 | 5.4 | 6.6 | 20.6 | 2.13 | 1.10 | 8.0 | 3.4 |
| 78°N | 81.5 | 13.8 | 22.7 | 6.7 | 5.8 | 22.0 | 4.64 | 2.98 | 8.4 | 4.4 |
| 54°N | 84 | 2.9 | 3.3 | 2.3 | 19.8 | 2.4 | 0.34 | 0.20 | 19.2 | 7.6 |
| 69°N | 84 | 2.2 | 4.2 | 1.0 | 21.6 | 5.0 | 0.96 | 0.70 | 20.0 | 9.8 |
| 78°N | 84 | 1.8 | 3.7 | 0.7 | 19.4 | 3.6 | 0.87 | 0.62 | 18.0 | 8.2 |

**Table 2.** Local time variation derived from daily data of $H_2O$ [ppmv] for two heights [km] at different latitudes for July 2009, see text for more details. Mean: mean $H_2O$ over a daily cycle ; Max: maximum $H_2O$ over a daily cycle ; Min: minimum $H_2O$ over a daily cycle; LT(Max): local time (LT) in hours of Max; LT(Min): local time (LT) in hours of Min; $A_{24}$: diurnal amplitude from a harmonic fit including 24h and 12h components; $A_{12}$: same but for the semidiurnal amplitude; $P_{24}$: diurnal phase of $A_{24}$ in LT hours of maximum; $P_{12}$: same but for semidiurnal phase.

components with an increase towards higher latitudes contrary to temperature amplitudes. The local time behavior of water vapor shows a pronounced maximum below PMC altitudes at 81.5 km during the morning between 5 and 7 LT. The phase position of maximum water vapor moves to some extent backwards in time in poleward direction, however, with a delay of approximately 3 hours when compared with temperature phases. Hence, both phase positions of low temperatures and large water vapor mixing ratios approximately coincide. For this reason we expect that the maximum strength of PMC formation should occur during morning hours as we will discuss in the next sections.

Generally, modeled PMC in MIMAS exist approximately poleward of 54°N where the degree of mean saturation $S$ is larger than unity. Saturation conditions are a combined effect of temperature, water vapor, ambient pressure, particle size and particle temperature. Figure 2 shows the saturation ratio $S$ at a fixed altitude of 82.7 km, which is the mean PMC altitude in the MIMAS simulation for the year 2009. The saturation ratio $S$ is approximated by $S = p_{H_2O}/p_\infty$ with equilibrium pressure $p_\infty$ and ambient partial pressure $p_{H_2O} = c(H_2O) \cdot p$, where $c(H_2O)$ is volume mixing ratio of water vapor and $p$ is pressure of air, for details see equations (1–3) in Berger and Lübken (2015) .

It turns out that most of the time supersaturation exists, only in the afternoon hours the saturation ratio falls below $S = 1$. The July average shows nearly permanently supersaturated conditions throughout the day. Note that the vertical extent of supersaturation areas increase polewards because of colder and higher mesopause conditions. In the following sections we will present model results of different PMC parameters and compare these with observational data.

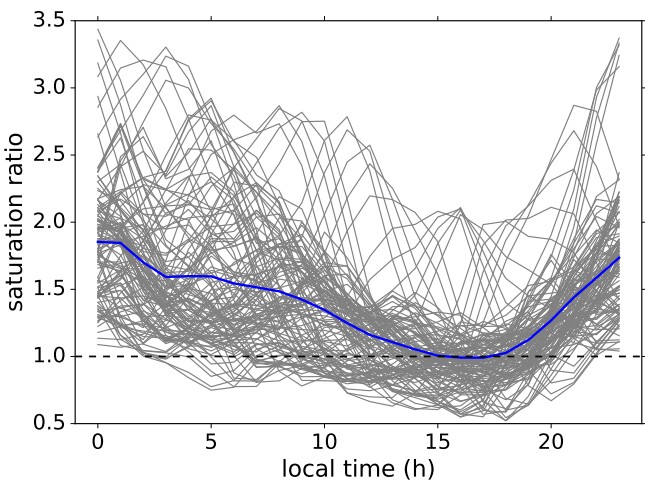

**Figure 2.** Hourly mean values of the saturation ratio ($S \approx p_{H_2O}/p_\infty$) in the latitude band $67^\circ$–$71^\circ$N for July 2009 at a fixed altitude of 82.7 km (mean PMC height) as function of local time. Grey lines show individual days and the blue line their mean.

## 3 Comparison of MIMAS backscatter model results with ALOMAR lidar observations

### 3.1 Seasonal variation of backscatter

During the northern hemispheric summer PMC typically occur from end of May until mid of August (e.g. Thomas and Olivero, 1989; Gadsden and Schröder, 1998; Hartogh et al., 2010; Hervig et al., 2013). At the core of the ice season in July, lowest temperatures near 130 K have been observed at mesopause altitudes near 88 km at $69^\circ$N (Lübken, 1999). Hence, we expect PMC most frequently and bright during July.

Figure 3 shows the mean seasonal variations of basic PMC parameters as calculated by MIMAS and observed by the Rayleigh/Mie/Raman(RMR)-lidar at the Arctic Lidar Observatory for Middle Atmosphere Research (ALOMAR), located at $69^\circ$N, $16^\circ$E (Fiedler et al., 2017). MIMAS results are limited to a latitudinal and longitudinal area of $67 - 71^\circ$N and $10 - 20^\circ$E to be close to the lidar position. We will use the volume backscatter coefficient of ice particles $\beta_{\max}$, in units of $10^{-10}\,\mathrm{m^{-1}sr^{-1}}$, as a measure for the cloud brightness. Both, model and observations cover the same time period of 11 years from 2003 to 2013. In order to take different cloud classes and the detection sensitivity of the lidar into account, we sort measurements and model results into different brightness ranges: $1 < \beta_{\max} < 4$ (faint clouds), $\beta_{\max} > 4$ (long-term detection limit of the lidar), and $\beta_{\max} > 13$ (strong clouds) (e.g. Fiedler et al., 2003; Baumgarten et al., 2008).

In order to convert the model output from MIMAS to specific lidar measurements, we apply spherical Mie-theory calculations to modeled ice particle distributions while taking into account the laser wavelength (532 nm) and scatter geometry ($180^\circ$). Finally, the transformed model results are sorted into brightness ranges. PMC brightness is proportional to the number of ice particles and depends approximately by the power of six on ice particle radius. For example, increasing the mean radius by only 25 % from 32 nm to 40 nm would result into a brightness change by a factor of 4. It is this high sensitivity of cloud

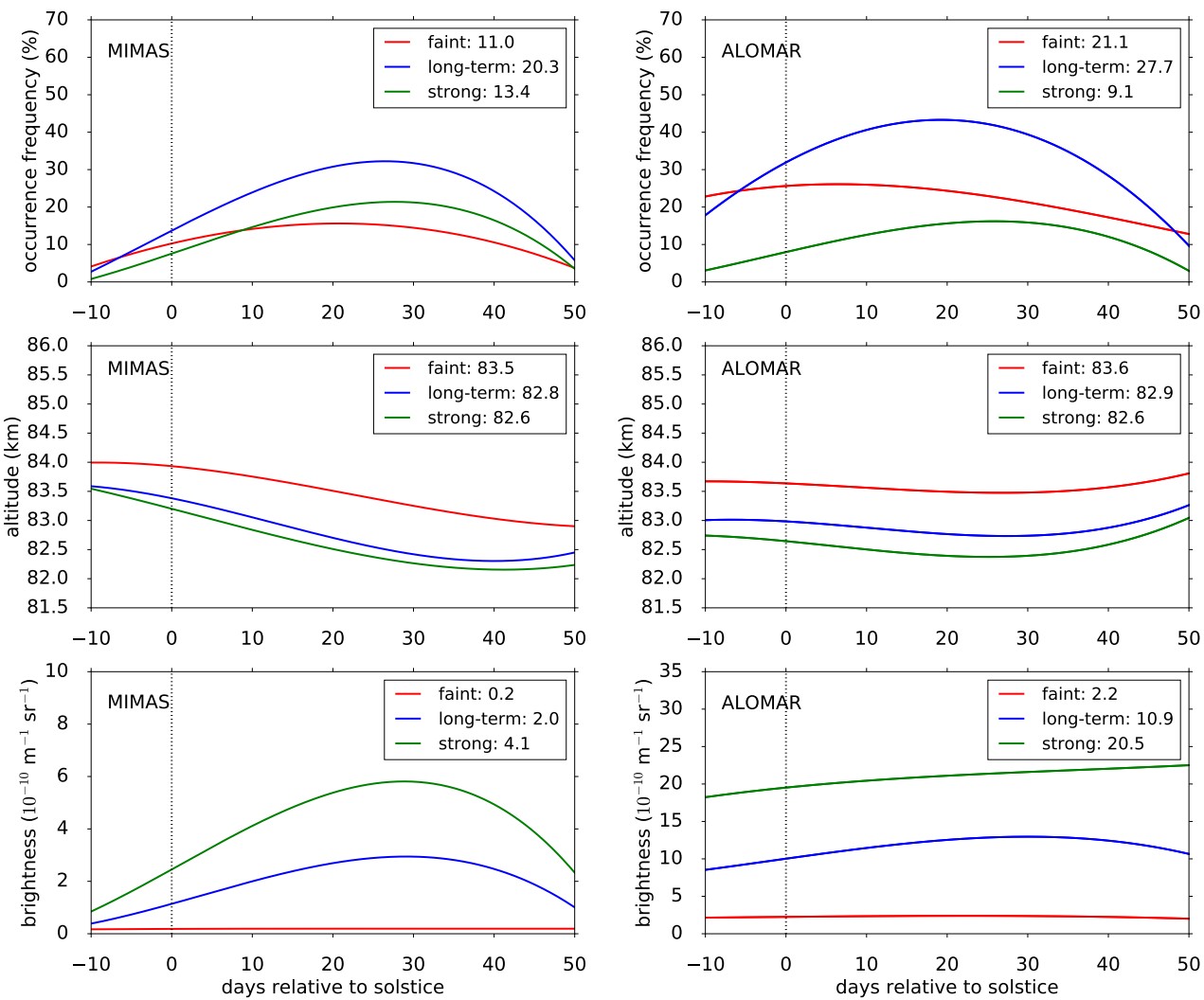

**Figure 3.** Mean seasonal variations of PMC occurrence frequency (upper panel), altitude (middle panel) and brightness $\beta_{max}$ (lower panel) between 2003 and 2013 at ALOMAR for faint (red), long-term (blue) and strong (green) clouds (for details see text). Left panels show model results for $67° - 71°$N, $10° - 20°$E, right panels show lidar observations from ALOMAR. The solid lines represent third-order polynomial fits based on daily means. Numbers in the Figure legends are seasonal mean values. Brightness ranges for cloud classes are scaled down by a factor of 4 for MIMAS results. Note the different scaling of the brightness axis for model and lidar data.

brightness to particle size that forms a hard benchmark for our complex ice model simulations. A small underestimation of the mean ice radius will dramatically decrease the brightness, on the other hand, a small overestimation will enhance the resulting backscatter signal by orders of magnitude. In order to match the mean occurrence frequencies of the lidar measurements we decreased the brightness ranges, defining the cloud classes, for the model results by a scaling factor of 4. Hence, the modeled occurrence frequencies contain a systematic bias. We think this deficiency is tolerable since our local time analysis relates to relative deviations from a mean. The scaling factor will only be used for the comparisons with lidar data in this and the following section.

The upper panels of Figure 3 show a general good agreement of modeled and observed PMC occurrence frequencies. We find maximum values in the long-term and strong cloud classes in mid July around days relative to solstice (DRS) $20-30$. Faint clouds observed by lidar occur earlier in the season than modeled faint clouds. This gives a hint that the model perhaps underestimates the microphysical process of nucleation in ice formation which essentially determines the frequency of weak PMC consisting of small ice particles. We note that ice nucleation in MIMAS is described by the concept of critical radius (Turco et al., 1982; Berger and von Zahn, 2002; Berger and Lübken, 2015).

The middle panels of Figure 3 show modeled and observed PMC altitudes which coincide quite well. Generally, weak PMC are at higher altitudes compared to strong PMC. This altitude separation is caused by two reasons. First, the sedimentation velocities of ice particles depend on their sizes. Weak PMC consist of ice particle distributions with smaller mean radii, typically in a range of 20 nm, whereas strong PMC consist of larger mean radii, e.g. 40 nm. As the sedimentation velocity increases with particle size (mass), larger particles can reach lower altitudes along their sedimentation path. Secondly, smaller ice particles start to sublimate at lower temperatures than larger ones due to the Kelvin effect. Thus, the negative vertical temperature gradient of the atmosphere causes smaller particles to sublimate at higher altitudes than larger particles. As a result larger ice particles, causing a higher brightness, are found at lower altitudes.

The lower panels of Figure 3 show modeled and observed PMC brightness. Here, the model results are calculated according to a given brightness range as an arithmetic mean of all brightness values matching the limits. Again, the model seems to underestimate begin and end of the season. The scaling factor for the brightness ranges leads to lower modeled brightness values in the different cloud classes. Hence, multiplying the modeled values with the scaling factor of 4 approximately reproduces the brightness values observed by lidar.

We summarize that the modeled seasonal distributions of occurrence, altitude and brightness are fairly consistent with the ALOMAR RMR-lidar observations, especially for July conditions. Therefore we will concentrate our discussion of model results in the following sections on this core period of the northern PMC season.

## 3.2 Local time variation of backscatter

PMC preferentially occur during morning hours which is attributed to thermal tides of background temperatures in the mesopause region (Fiedler et al., 2011). In order to validate the structure of local time variations in MIMAS we compare our model results to observations by the RMR-lidar at ALOMAR and to instruments on-board the AIM satellite. For comparison to lidar data we will apply a scaling factor of 4 regarding the brightness ranges, defining the cloud classes, as described in the previous section.

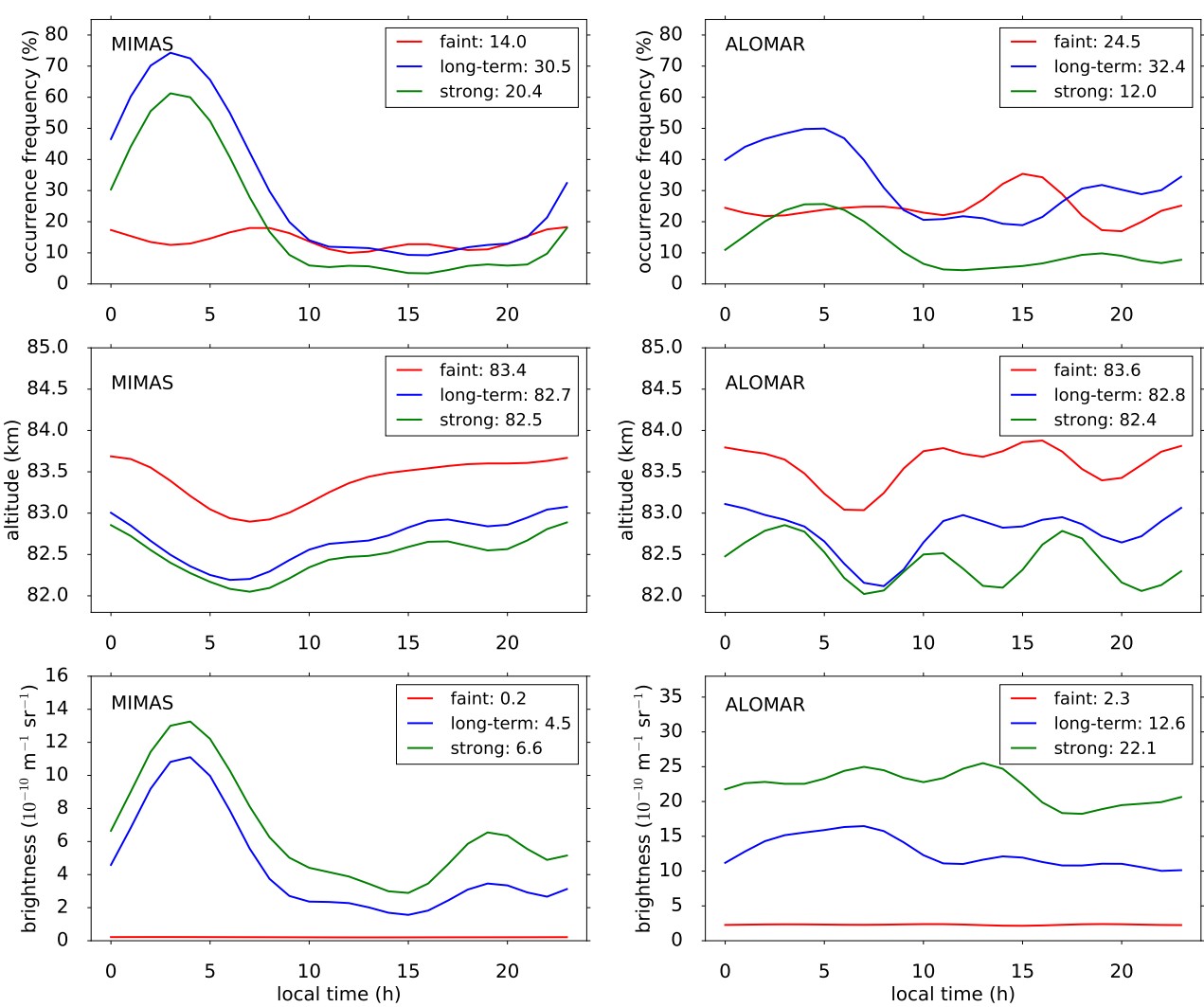

**Figure 4.** Mean local time variations of PMC occurrence frequency (upper panel), altitude (middle panel) and brightness $\beta_{\mathrm{max}}$ (lower panel) for July in the period from 2003 to 2013 at ALOMAR for faint (red), long-term (blue) and strong (green) clouds (for details see text). Left panels show model results for $67° - 71°$N, $10° - 20°$E, right panels show lidar observations from ALOMAR. The lines represent the sum of four harmonic fits using periods of 24 h, 12 h, 8 h, and 6 h to hourly mean values. Numbers in the Figure legends are daily mean values. Brightness ranges for cloud classes are scaled down by a factor of 4 for MIMAS results. Note the different scaling of the brightness axis for model and lidar data.

As discussed above we will concentrate on the core period of the northern PMC season and will use only July data (31 days x 24 h) from MIMAS simulations for the PMC seasons 2003 – 2013. Tidal structures in the LIMA model have been discussed earlier by Herbort et al. (2007) and Fiedler et al. (2011).

Figure 4 shows the variation of PMC occurrence frequency, altitude, and brightness throughout the day for the integrated
data set of July 2003 – 2013 and brightness classes as defined above. The curves are superpositions of four harmonic functions with periods of 24 h, 12 h, 8 h, and 6 h, which are fitted to hourly mean values as described in Fiedler et al. (2017). The geographic range is again restricted to the area around ALOMAR. We find pronounced and persistent features which indicate a strong influence of tides on PMC parameters. The occurrence frequency variation over a day is largest for strong clouds both in MIMAS and observations. Like in the observations, the model results show highest cloud occurrence during the morning
hours. The local time dependencies of altitude and brightness are anti-correlated, i.e. on average ice clouds of higher brightness are found at lower altitudes. In general, a predominant diurnal oscillation exists in agreement with the lidar observations. The lidar observations show additionally semidiurnal variations in all three PMC parameters, which seems to some extent underestimated by the model. Contrary, the modeled brightness shows a clear peak in the morning hours around 4 LT which is absent in the observations.
In order to investigate these different structures we calculated the ratios of diurnal to semidiurnal tidal amplitudes ($A_{24}/A_{12}$). The values in Table 1 show that both model and lidar fits have nearly the same amplitude ratios for a number of cloud parameter and class combinations. For example, for the long-term brightness the ratios are 1.82 (model) and 1.88 (lidar), meaning that tidal modes are very similar in both data sets. Thus the phase differences of modeled and observed data, especially for the semidiurnal modes, (not shown here) are mostly responsible for the differences visible in Figure 4. The superposition of
diurnal and semidiurnal tidal modes yields a stronger morning peak in the modeled compared to the observed brightness.

In summary, observed local time variations of PMC occurrence and brightness at ALOMAR are fairly well reproduced by MIMAS.

| | **MIMAS** | | | **RMR-lidar** | | |
| --- | --- | --- | --- | --- | --- | --- |
| | OF | altitude | brightness | OF | altitude | brightness |
| faint | 1.40 | 2.25 | 5.71 | 0.89 | 0.67 | 0.71 |
| long-term | **2.45** | 2.36 | **1.82** | **2.51** | 0.77 | **1.88** |
| strong | **2.00** | 1.75 | **1.96** | **1.59** | 3.02 | **2.44** |

**Table 3.** Ratio of diurnal to semidiurnal amplitudes ($A_{24}/A_{12}$) of harmonic fits to the modeled and observed occurrence frequency (OF), altitude, and brightness. The values are calculated for different cloud classes (for details see text) for July months in the period from 2003 to 2013 at ALOMAR according to Figure 4. Bold numbers mark values that agree within the relative uncertainty of about 15 % (confidence level of 95 %).

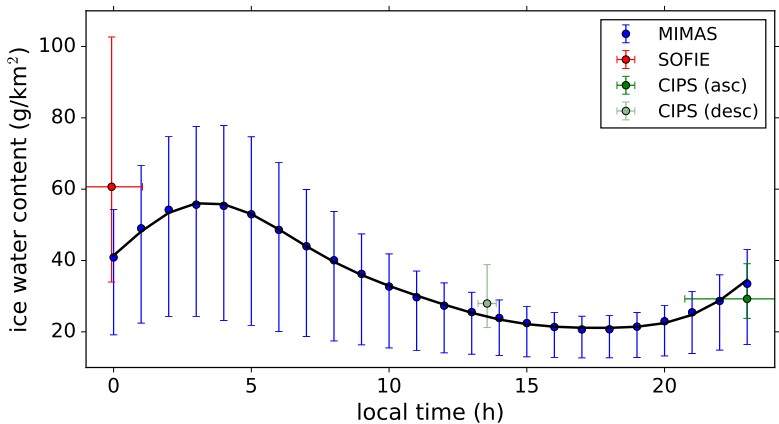

**Figure 5.** Hourly median values of IWC from $2007 - 2013$ (July) for $67° - 71°$N and IWC threshold of $10\,\text{g/km}^2$ as a function of local time. The vertical bars represent the lower and upper quartile of the data. The black curve is a harmonic fit to the data with periods of 24 h, 12 h, and 8 h. Data from AIM satellite instruments including uncertainties for the same time range: SOFIE V1.3 (red - from http://sofie.gats-inc.com/sofie/index.php) and CIPS Level 3c (green - from http://lasp.colorado.edu/aim/download-data-pmc.php) for ascending and descending nodes.

## 4 Comparison of MIMAS ice water content model results with AIM satellite observations

Comparison of PMC brightness values between different instruments is affected by observational constraints, e.g. viewing geometry, lighting conditions, temporal overlap and wavelength. Stevens et al. (2005) suggested that integrated ice mass has the advantage to be less dependent on instrumental setups and thus should be more robust to be used for PMC comparisons. Therefore we present in this section model results of ice water content (IWC) which are calculated from the integrated ice mass density over the total vertical ice column. We analyze the time period $2007 - 2013$ to cover the time range of the SOFIE instrument on-board the AIM satellite. The IWC is calculated from all longitudes in the latitude band $67° - 71°$N. In order to resolve tidal structures we subdivide each latitudinal circle into 120 longitudinal segments and sort the model data according to actual local times at all segments. This method yields a total of 4 latitudes times 120 longitudes times 31 days times 24 h of values for July conditions. Finally, we average all IWC values corresponding to a certain local time with a local time resolution of one hour per day. The probability density distributions of all these IWC values show to a high degree an exponential behavior. Therefore we calculate two different averages (median and arithmetic mean), in order to characterize a mean ice water content as a function of local time during July.

In Figure 5 we compare our IWC model results in terms of median values with measurements from the CIPS and SOFIE instruments on-board the AIM satellite for the latitude band $67° - 71°$N. The AIM satellite operates in a sun-synchronous orbit, hence only limited local times are available (Russell et al., 2009). For comparison with model results we take the different sensitivities of the two AIM instruments (SOFIE, CIPS) into account. The detection threshold for SOFIE is given as $0.5\,\text{g/km}^2$ (Hervig et al., 2009a). In contrast to SOFIE, the CIPS instrument is less sensitive allowing only IWC events

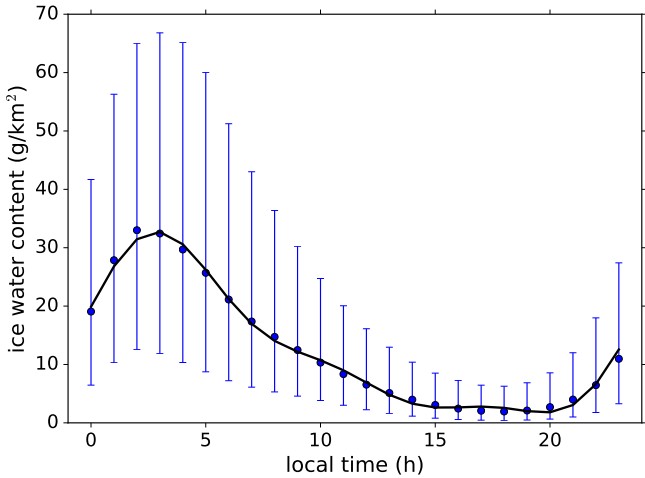

**Figure 6.** Hourly median values of IWC from $2007 - 2013$ (July) for $67° - 71°$N as a function of solar local time. No threshold has been applied, IWC values of zero (no PMC) are included. The vertical bars represent the lower and upper quartile of the data. The black curve is a harmonic fit to the data with periods of 24 h, 12 h, and 8 h.

larger than $10 \, \mathrm{g/km^2}$ to be detectable (Lumpe et al., 2013). Hence all IWC data sets (MIMAS, SOFIE, CIPS) are limited to this threshold. We find a good agreement between model results and the data points from SOFIE and CIPS inside the error bars. Generally, the modeled IWC has maximum values in the early morning hours between 1 and 4 LT and lowest values between 16 and 20 LT. On average the IWC varies by a factor of about two during a day. Interestingly, comparing SOFIE with CIPS data, the CIPS observation at 23 LT does not match the SOFIE point for midnight conditions. There is a substantial deviation between these values of (SOFIE: $60 \, \mathrm{g/km^2}$, CIPS: $30 \, \mathrm{g/km^2}$) which might be due to some uncertainties in the CIPS threshold. The MIMAS value of $40 \, \mathrm{g/km^2}$ is right in between the two different satellite observations. Nevertheless, all three data points coincide within their error bars.

We summarize that the MIMAS model results of PMC ice water content are compatible to a high degree with the satellite observations.

Figure 6 shows again the IWC local time variation for the latitude band $67° - 71°$N, but now without any threshold which means that IWC has been frequency weighted and IWC values of zero (no PMC) are included. This yields an IWC variation over day by a factor of ten compared to the factor two when considering the threshold used in Figure 5. The factor of ten derived from frequency weighted IWC is consistent with model results reported by Stevens et al. (2010) (see their Figure 7). Hence, the strength of local time variations is sensitive to the IWC threshold, meaning that larger thresholds induce smaller local time variations, see discussion in section 6. The times of IWC maxima and minima are close to those of occurrence frequency and the brightness as shown in Figure 4. We find the harmonic fit to be highly correlated to the median values (correlation coefficient of 0.99), meaning that the local time behavior of IWC medians is almost perfectly represented by the

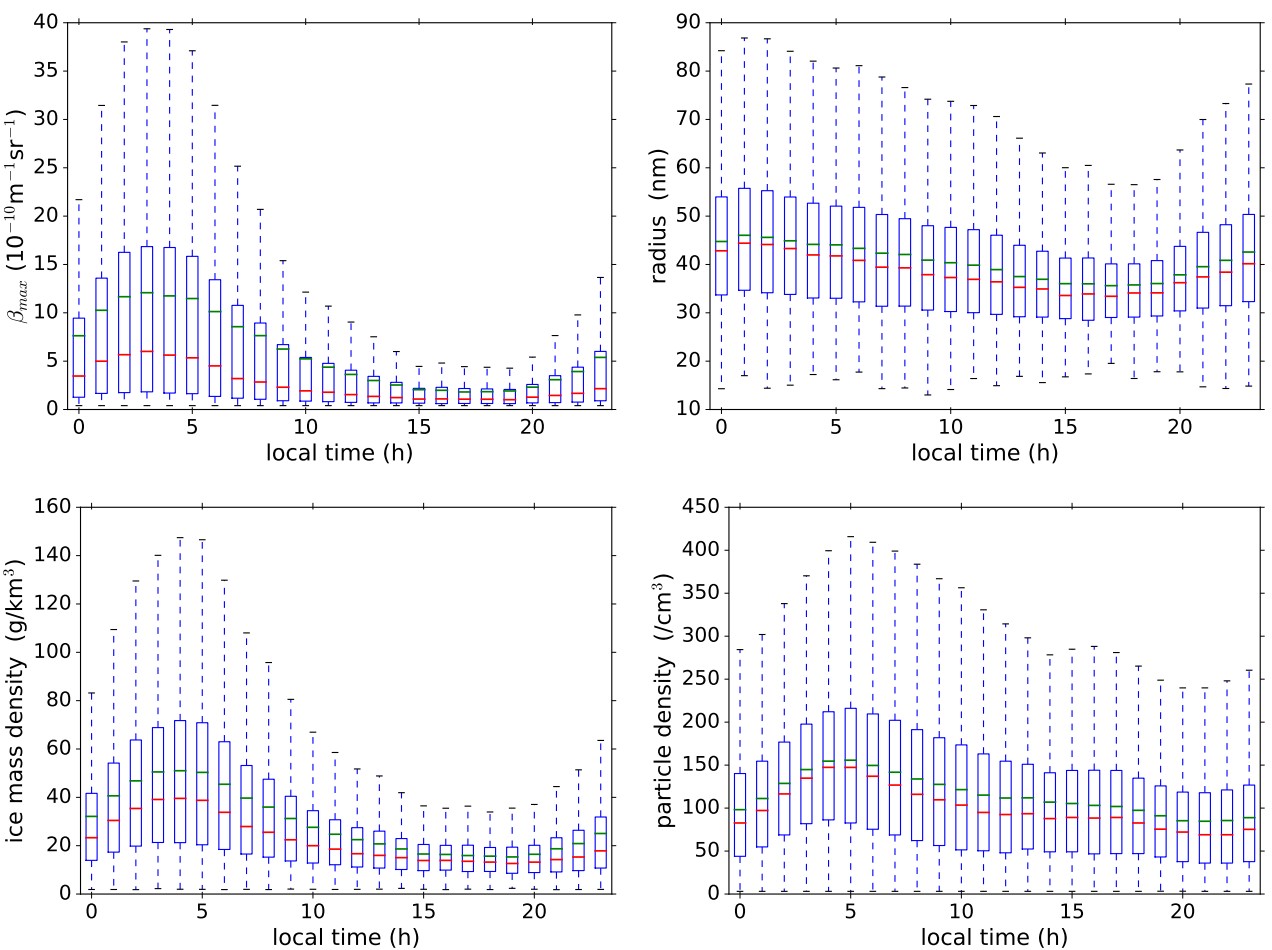

**Figure 7.** Ice parameters at $67° - 71°$N calculated from MIMAS simulations of all July months $2003 - 2013$ for the altitude range near 83 km where $\beta_{\max} > 0.4$. Upper panels: brightness and ice particle radius. Lower panels: ice mass density and particle number density. The boxes represent lower and the upper quartiles, median (red line), and arithmetic mean (green line). The dashed vertical bars indicate the minimum and maximum values.

three harmonics of 24, 12, and 8 hours. The fit is dominated by the diurnal and semidiurnal mode, the terdiurnal mode is of minor importance. The amplitude ratios are $A_{24}/A_{12} = 2.66$ and $A_{24}/A_8 = 5.84$.

## 5   Local time behavior of ice particle radius, number, and ice mass density

In the previous sections we compared MIMAS simulations of backscatter and ice water content with observations in order to
5   show that MIMAS provides realistic model results. Now we investigate in more detail the local time variations in different ice parameters as ice particle number density, ice particle radius, and ice mass density in comparison to backscatter.

Our model simulations of PMC show that the number of ice particles is largest at mesopause altitudes between 86 and 89 km where the highest chance of nucleation is found. This altitude region serves as a reservoir of small ice particles. Then, slightly below mesopause altitudes the MIMAS model predicts the largest number density of ice particles to fall in the range 500 to 1500 cm$^{-3}$ ($67° - 71°$N). The mean radius of ice particles stays generally below 15 nm, which is usually too small to produce significant lidar backscatter signals. Due to random diffusive transport processes a fraction of these small ice particles experiences enhanced growing. The increase in particle mass enhances downward sedimentation. Towards lower altitudes the amount of free background water molecules increases exponentially since air density increases exponentially. During their downward sedimentation path the growth of ice is stimulated until an ice particle reaches an altitude where supersaturated background conditions change into undersaturation. This is the height where the radius of ice particles maximizes and thus highest ice mass densities and largest backscatter signals occur.

In Figure 7 we present backscatter, mean ice radius, number density, and ice mass density at the altitude of maximum backscatter signal, assuming a threshold of $\beta_{\max} > 0.4$, for the latitude band $67° - 71°$N during July. The plots show both median and arithmetic mean values. Median and arithmetic mean are generally different which indicates that the underlying distributions are not symmetric.

Mean ice radii vary between 35 and 45 nm. These numbers are in good agreement with AIM-SOFIE observations which also indicate ice radii of 35–40 nm (Hervig et al., 2009a). Mean ice particle densities fall in the range 80 to 150 cm$^{-3}$, which agrees with results from lidar observations (Baumgarten et al., 2008) and satellite measurements (Hervig et al., 2009a). Similar to ice radii, the mean ice mass density increases from the heights below the mesopause downward with mean values about 30 g/km$^3$ at PMC heights. It is interesting to note that the low altitude boundary of the backscatter at 69°N as simulated by MIMAS indicates a temperature of 150 K which agrees well with the observed temperature of $150 \pm 2$ K for the low altitude boundary of NLCs (Lübken et al., 1996).

Investigating the local time dependence of ice parameters we find that the ice number density maximizes in the morning hours between 3 and 5 LT, which corresponds with the maxima of ice mass density and $\beta_{\max}$. The mean radius shows a smaller variation with local time and no pronounced maximum during in the morning. This indicates that the local time behavior of ice mass density is mainly determined by the number of ice particles and less by the ice particle radius. Our model results are confirmed by AIM observations which show that an increase in ice mass is significantly correlated with increasing number densities and less correlated with the size of ice particles (Hervig et al., 2009b). We mention that model calculations performed with the 1–d ice model CARMA show some controversial results, meaning that particle number density has no effect on ice mass and brightness (Megner, 2011).

In MIMAS local time dependencies in ice parameters are mainly forced by tidal variations in background temperature and water vapor as has been discussed in section 2.2. Local time dependence of brightness in terms of $\beta_{\max}$ with a diurnal maximum near 4 LT follow nicely the temperature structure with a diurnal minimum at 4–5 LT, see Table 1. In addition, we find the maximum water vapor to occur between 6 and 7 LT and hence about 2–3 hours after the brightness maximum, cf. Figure 7 and Table 2. We conclude that the local time phases in temperature and water vapor are the main drivers to determine the phase structure in ice parameters.

| Latitude band | $\frac{A_{24}}{A_{12}}$ | $\frac{A_{24}}{A_8}$ | max/min |
|---|---|---|---|
| $61° - 65°$N | 7.6 | 6.0 | 12.6 |
| $67° - 71°$N | 2.2 | 4.1 | 18.3 |
| $73° - 77°$N | 2.1 | 4.8 | 10.4 |
| $79° - 83°$N | 2.7 | 7.8 | 6.9 |

**Table 4.** Ratios of IWC tidal amplitudes for July 2009 and different latitudes bands. No threshold has been applied, IWC values of zero (no PMC) are included. The ratios of maximum to minimum IWC indicate the variability throughout the day. For details see text.

# 6 Latitudinal variations of local time dependence for ice water content

Our numerical simulations indicate that the local time variations of PMC are subject to significant latitudinal dependencies. Figure 8 shows modeled IWC values over latitude for selected local times in July $2007 - 2013$. No threshold was applied and IWC values had been frequency weigthed so that median values include 'zero' PMC events. While at 6 LT IWC increases nearly linearly from $60°$N to $84°$N, the slopes are quite different throughout the rest of the day. This indicates that the phase of the local time behavior changes with latitude. As an example, the time of IWC maximum changes from the morning hours at mid latitudes to midnight hours at high latitudes. Figure 9 shows this phase variation in more detail for different latitude bands. It turns out that (1) the amplitude of the local time dependence increases in absolute IWC values towards the pole, (2) the ratio of maximum to minimum IWC decreases towards the pole (see Table 4), and (3) a slight phase shift can be seen with decreasing latitude: the IWC maximum around midnight near $81°$N moves forward in time to 4 LT near $63°$N.

IWC median values at mid latitudes are much smaller (about 100 times) than those at high latitudes. Therefore we also use the ratio of daily maximum to minimum IWC values as an additional indicator for local time variations, see Table 4. Please note that the ratios are calculated from median IWC values without any lower threshold, hence the occurrence frequency has a large influence on the median value. This is in particular important at the lowest latitude band ($61° - 65°$N) where rather small PMC occurrence frequencies are modeled. E.g., assuming an IWC threshold of 5 g/km$^2$, the PMC occurrence frequency at this latitude band is only in the order of $5 - 10$ % during July whereas moving poleward it increases to about 50 % at $67° - 71°$N and 100 % at $79° - 83°$N. For this reason results for the lowest latitude band ($61° - 65°$N) in Table 4 include enhanced uncertainties.

Table 4 also includes tidal amplitude ratios obtained from fitting of 24 h, 12 h, and 8 h harmonic components. We find that for the three highest latitude bands the diurnal component is generally about two times larger than the semidiurnal component. This ratio $A_{24}/A_{12}$ seems to be fairly independent of latitude. There exists a terdiurnal component with a strength of about 20 % which decreases in poleward direction. On average the ratio of daily maximum to minimum IWC values is about 10 and decreases towards the pole.

Now we investigate the local time structure of IWC and its latitudinal dependence in terms of different IWC thresholds. In the following, IWC data are not frequency weighted. Additionally we extent the time period to range from 1979 to 2013, thereby presenting a 35-y climatology of daily fluctuations which aims to describe mean local time variations. Such specifications might be useful for satellite data analysis in order to perform local time corrections. The results are shown in Table 5. The

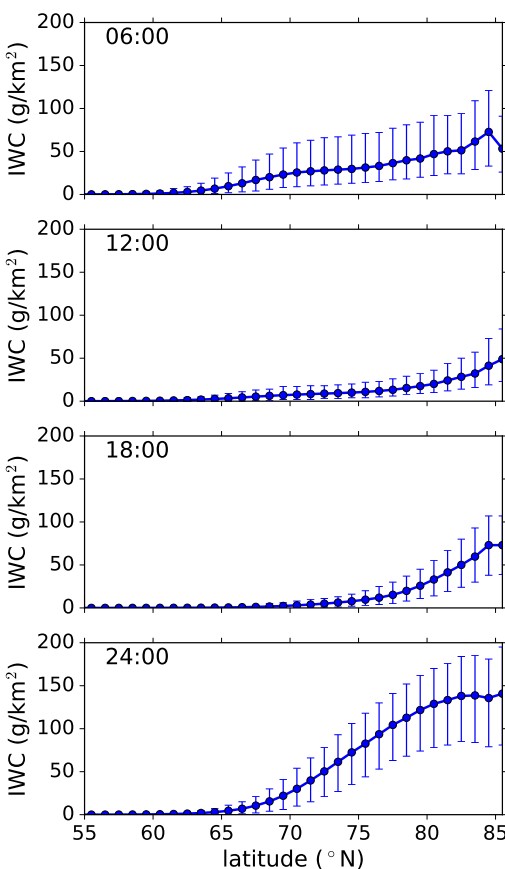

**Figure 8.** Median IWC values for July 2007 – 2013 as function of latitude for different local times. No threshold has been applied, IWC values of zero (no PMC) are included. The vertical bars represent the lower and upper quartile of the data.

modeled IWC data have been calculated over three latitude bands used in SBUV trend analysis and for three thresholds with $IWC > 0$ g/km$^2$, $IWC > 10$ g/km$^2$, and $IWC > 40$ g/km$^2$. The latter threshold was used in SBUV trend analyses by DeLand and Thomas (2015) and Hervig and Stevens (2014). Both absolute means and absolute local time variations, expressed here as difference between maximum and minimum value, increase towards the pole. We find the ratio of maximum to minimum values, a measure for the relative IWC local time variation, to increase poleward too. Additionally, IWC ratios decrease with higher thresholds, e.g. at latitudes $64° – 74°$N from 6.6 ($IWC > 0$) to 2.4 ($IWC > 10$) and 1.7 ($IWC > 40$), see Table 5 (7th column).

Maximum values of IWC occur in general during the early morning hours whereas minimum values are present in the afternoon hours. Local times of IWC maximum and minimum are independent of the selected threshold. There exists a time shift in latitudinal direction, e.g. at polar latitudes $74° – 82°$N the maximum occurs at 2 LT for $IWC > 40$ g/km$^2$ whereas at mid latitudes $50° – 64°$N it is shifted forward in time to 8 LT. Recently, Stevens et al. (2017) reported about model results

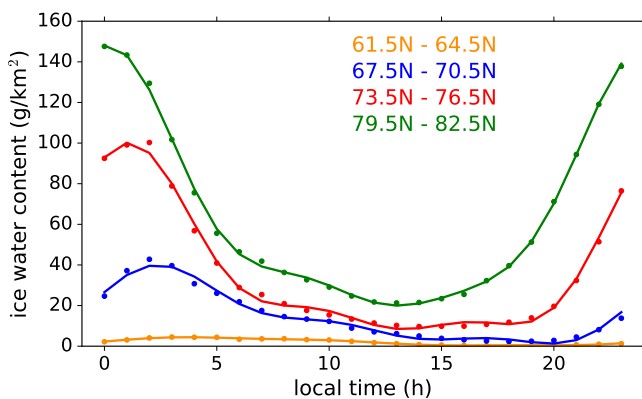

**Figure 9.** Diurnal variation of hourly median IWC values for July 2007 – 2013 for different latitude bands. No threshold has been applied, IWC values of zero (no PMC) are included. Dots indicate the data and solid lines are harmonic fits using periods of 24, 12, 8 h.

of PMC IWC calculations with the NOGAPS-ALPHA model using a 1-d bulk ice model (Hervig et al., 2009b). The authors show that the IWC is largest at highest latitudes and yields a morning peak between 5 and 7 LT and a late afternoon minimum equatorward of 80°N regardless of threshold. Diurnally averaged IWC values (threshold of 40 g/km$^2$) are near 100 g/km$^2$ and consistent with those calculated by MIMAS. NOGAPS-ALPHA results of IWC over a diurnal cycle show at 68°N a ratio
between IWC maximum and minimum of about 1.5 for a threshold of 40 (see Figure 6a,b in Stevens et al. (2017)) similar to a ratio of 1.7 from MIMAS calculations. Concurrently, absolute IWC local time variations in NOGAPS-ALPHA increase towards higher latitudes and are threshold dependent. Again, these features are confirmed by MIMAS.

Lidar observations of daily variations of mid-latitude NLC (54°N, Kühlungsborn, Germany) show highest rates at 5–6 LT which is similar to our model result (Gerding et al., 2013). On the other hand, DeLand et al. (2011) published local time
observations by the Aura OMI (Ozone Monitoring Instrument) satellite instrument which indicates maximum frequency and albedo values at approximately 9-10 h LT at 70°N for the NH 2007 season, with a smaller amplitude and a slight phase shift to ∼8 h LT at higher latitudes. Hence, model results from MIMAS deviate to some extent from these satellite measurements for 2007. Here we refer to some year-to-year variations of phases in MIMAS (not shown here) which might explain to some extent these differences.

As shown in section 2.2, phase positions of minimum temperature at PMC altitudes move to some extent during early morning hours backwards in time in poleward direction. Also the phase of the daily water vapor maximum tends to follow this time shift. We conclude that both temperature and water vapor phases cause the general early morning hour structure in IWC and its shift towards higher latitudes.

Generally, the time difference between IWC maximum and minimum is approximately constant with 12 hours at all latitudes
and for all three thresholds. This indicates that a tidal decomposition of daily data reveals the significant role of the diurnal tidal oscillation. Indeed, all daily time series of IWC are approximated to a high degree by harmonic fits of a dominant 24 h and a minor 12 h component, the ratio $A_{24}$ /$A_{12}$ varies between 4.5 and 8.8. Hence, semidiurnal fluctuations in IWC are of

| Latitude band | Threshold | Mean | Max | Min | Max-Min | Max/Min | LT(Max) | LT(Min) | $A_{24}$ | $A_{12}$ | $P_{24}$ | $P_{12}$ |
|---|---|---|---|---|---|---|---|---|---|---|---|---|
| $50°-64°$N | 0 | 2.9 | 4.6 | 1.5 | 3.1 | 3.1 | 8 | 21 | 1.5 | 0.2 | 7 | 7 |
| $64°-74°$N | 0 | 17.1 | 32.8 | 5.0 | 27.8 | 6.6 | 3 | 19 | 14.6 | 2.2 | 5 | 4 |
| $74°-82°$N | 0 | 48.4 | 102.3 | 16.0 | 86.3 | 6.4 | 2 | 15 | 42.4 | 9.4 | 2 | 2 |
| $50°-64°$N | 10 | 18.9 | 24.0 | 14.6 | 9.4 | 1.6 | 8 | 20 | 4.4 | 0.5 | 7 | 5 |
| $64°-74°$N | 10 | 36.6 | 53.4 | 21.8 | 31.6 | 2.4 | 3 | 19 | 16.1 | 2.0 | 5 | 3 |
| $74°-82°$N | 10 | 61.1 | 109.8 | 30.6 | 79.2 | 3.6 | 2 | 15 | 38.7 | 8.3 | 2 | 1 |
| $50°-64°$N | 40 | 47.6 | 55.5 | 41.4 | 14.1 | 1.3 | 8 | 20 | 7.0 | 0.9 | 7 | 6 |
| $64°-74°$N | 40 | 69.7 | 86.9 | 52.1 | 34.8 | 1.7 | 3 | 19 | 17.2 | 2.0 | 5 | 2 |
| $74°-82°$N | 40 | 92.6 | 132.5 | 63.9 | 68.6 | 2.1 | 2 | 15 | 34.6 | 5.7 | 3 | 2 |

**Table 5.** Climatology of local time variations of IWC in units of g/km$^2$ for three thresholds (IWC $> 0$, IWC $> 10$, and IWC $> 40$) at different latitudes bands for the period July 1979–2013. Mean: mean daily IWC over a daily cycle; Max: maximum IWC over a daily cycle ; Min: minimum IWC over a daily cycle; Max-Min: difference between maximum and minimum IWC; Max/Min: ratio between maximum and minimum IWC; LT(Max): local time (LT) in hours of Max; LT(Min): local time (LT) in hours of Min; $A_{24}$: diurnal amplitude from a harmonic fit including 24 h and 12 h components; $A_{12}$: same but for the semidiurnal amplitude; $P_{24}$: diurnal phase of $A_{24}$ in LT hours of maximum; $P_{12}$: same but for semidiurnal phase.

minor importance which again is explained by small semidiurnal tidal amplitudes in temperature and water vapor. We note that terdiurnal tidal components are also present. But on average 8 h amplitudes are in the order of 20 % of 12 h amplitudes and therefore have a negligible impact.

5    We summarize that these results highlight the importance of taking tidal PMC variations into account when compiling data sets which are distributed over latitude and local time. It turns out that for IWC (1) local time variations depend on threshold conditions, e.g. relative local time variations decrease with larger thresholds; (2) local time variations depend on latitude, e.g. absolute local time variations increase towards the pole; (3) a phase shift exists towards the pole which is independent of the threshold value, e.g. the IWC maximum moves backward in time from 8 LT at mid latitudes to 2 LT at high latitudes. The IWC local time behaviour presumably exhibits year-to-year as well as long-term variability which may effect the 35-y mean state

10   given in Table 5. However, this needs more detailed investigations and will be subject of future work.

## 7    Conclusions

In this paper we presented a detailed investigation of tidal effects on PMC occurrence, altitude, brightness and microphysical properties of ice particles as calculated by the MIMAS model. As already discussed in several publications, the interpretation of PMC observations requires a careful treatment of the local time of the observations even for the investigation of long-term

15   records (Fiedler et al., 2011; DeLand and Thomas, 2015; Stevens et al., 2017). We have compared our results to observations by ground-based lidar as well as satellite instruments and find a good agreement when taking into account instrumental sensitivity

and local time of observations. MIMAS reproduces the local time variations seen by lidar especially well in the core of the PMC season. PMC simulations for ALOMAR show in the latitude range $67° - 71°$N brightness variations throughout the day up to a factor of 7, while the occurrence frequency varies by a factor of 2 to 16 for faint and strong clouds, respectively. At the peak of the PMC layer the mean ice particle radius varies from 35 to 45 nm and the mean number density from 80 to 150 cm$^{-3}$ throughout the day. All quantities show the maximum around a local time of $3\pm2$ h. At the same latitude band the time of maximum IWC is about 3 LT and the minimum is found around 18 LT. Without thresholding the data, hourly IWC median values vary by a factor of 10 throughout the diurnal cycle. In general diurnal and semidiurnal tides in temperature and water vapor contribute to the tidal behavior of PMC parameters whereas terdiurnal tidal structures are of minor importance.

Our analysis shows that the local time dependence becomes most evident when concentrating on one single season. When limiting the analysis to the season 2009 we find that local time variations of temperature at $69°$N are in a range of 6 K near 83 km altitude. At sublimation altitudes (near 81.5 km) the water vapor variation is about 7 ppmv. The variation in water vapor leads to a change of the saturation ratio from about 1.8 around midnight to 1 in the afternoon.

We calculated a climatology of IWC local time variations from a 35-y average from 1979 to 2013 for different thresholds and latitude bands, which might be useful for satellite data analysis in order to perform local time corrections. Local time variations are found to depend on latitude and threshold conditions. For the latitude band $64°$–$74°$N and a threshold of IWC $> 0$ g/km$^2$ IWC maximum and minimum values occur around 3 LT and 19 LT, respectively, with a ratio maximum to minimum of 6.6. For a threshold of IWC $> 40$ g/km$^2$ the local times for maximum and minimum are identical, but the ratio changes to 1.7. A phase shift exists for the IWC local time behavior towards the pole, which is independent of the threshold value. We find the absolute IWC local time variation to generally increase with latitude. Furthermore, the IWC maximum moves backward in time from 8 LT at mid latitudes to 2 LT at high latitudes.

It should be noted that gravity waves could mask the influence of tides especially for the terdiurnal component. Gravity waves are partly included in the MIMAS model, but a detailed investigation regarding their effects on the tidal behavior of PMC is beyond the scope of this paper. However, we expect that the latitudinal variations of tidal amplitudes are robust and will help interpreting long-term observations with varying latitudes and fixed or variable local times.

*Acknowledgements.* We appreciate the financial support from the German BMBF for the ROMIC/TIMA project. This research was supported by the European Union's Horizon 2020 Research and Innovation program under grant agreement No 653980. We thank the AIM community for providing us with SOFIE and CIPS data that are available online at http://sofie.gats-inc.com/sofie/index.php and http://lasp.colorado.edu/aim/download-data.php, respectively. The European Centre for Medium-Range Weather Forecasts (ECMWF) is gratefully acknowledged for providing ERA-40 and operational analysis data.

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
