# Peer review of "Local Time Dependence of Polar Mesospheric Clouds: A model study"

_Atmospheric Chemistry and Physics, 2017_

## Referee Comment (RC1) · Anonymous Referee #1 · 3 Nov 2017

SUMMARY

This paper presents an analysis of local time variations in polar mesospheric cloud (PMC) properties using a 3-D atmospheric model (MIMAS). The results are compared to local time variations derived from lidar data at a single location (ALOMAR in Norway), as well as zonal average results from the SOFIE and CIPS instruments on the AIM satellite. MIMAS also calculates many parameters describing the background atmosphere [e.g. temperature, water vapor, ice particle radius] that are examined for their contributions to local time variations.

GENERAL COMMENT: For better or worse, we may never get a satellite measurement of PMCs with simultaneous SOFIE-level sensitivity and comprehensive global coverage. So if these model results are to be validated against satellite data, I think

that presenting curves based on some of those higher thresholds would be quite valuable. The authors might wish to primarily use qualitative statements in the main paper, and provide extra figures in an appendix or on-line supplement (since this paper is a "model study"). But since there is the possibility of non-linear behavior in going from no threshold in IWC to a SBUV-type threshold (for example), I think that providing such information somewhere would help the acceptance of the large variations shown in some aspects of this analysis.

This paper is well-written. Some suggestions and comments related to specific items are provided below.

SPECIFIC COMMENTS

1. p. 1, lines 23-24: So the relative strength of these components (where both are present) is actually a guide to lower atmosphere structure? This is relevant to comment #10.

2. p. 2, lines 13-14: Please clarify that this limitation is due to local time sampling, not spatial coverage.

3. p. 2, lines 15-18: Please note also that in contrast to the previous statement, the restricted spatial coverage of lidar data presents a limitation in terms of how well results from any single location can be generalized to other locations (both latitude and longitude).

4. p. 5, lines 18-22: This seems like a reasonable choice because the model can probably form clouds more easily. However, the next paragraph (e.g. lines 25-27) seems to give a different result. Since local time variations are a perturbation on existing clouds, they presumably indicate increased sensitivity to the effectiveness of formation mechanisms. This sensitivity should be addressed later.

5. p. 6, lines 12-13: Please connect this concept to the ideas mentioned on the bottom of page 1 regarding how mesospheric thermal variations are being forced.

6. p. 6, lines 23-25: The magnitude of the model variations is significantly larger than the satellite results. Stevens et al. [2017; J. Geophys. Res. Atmos. 122, doi:10.1002/2016JD025349, Section 3.1] discuss the potential differences depending on whether "zero" values are included in averages, but these differences seem large even when that issue is considered.

7. p. 6, lines 28-30: These results can be related to the diurnal and semi-diurnal mechanisms discussed on p. 1.

8. p. 9, line 5: This variation in IWC is still much larger than the fit to the SBUV data ($\sim$15-20% p-p), even given the uncertainty in that result because of the nature of the local time coverage. This makes me question the strength of the statement "compatible to a high degree" on lines 10-11.

9. p. 9, lines 14-15: See "General Comment" at the beginning of this review. Does a threshold of 40 g/kmˆ2 reduce the local time variation down to the magnitude shown in DeLand and Thomas [2015]?

10. p. 10, line 1: The physical arguments presented on p. 1 imply that large ratio values of A24/A12, as listed here, mean that tropospheric forcing of tidal variations is much more important for PMC formation and growth than stratospheric forcing. Is this an appropriate statement?

11. p. 12, lines 4-5: This result seems surprising given the discussion of high sensitivity to particle radius on p. 5, lines 13-15. Even a few nm matters with an rˆ6 dependence. Comments?

12. p. 12, lines 15-16: This seems like a significant variation in PMC altitude, considering the small magnitude of quoted long-term variations in z_PMC by Berger and Lubken [2015].

13. p. 13, lines 16-17: What happens with a higher IWC threshold? DeLand et al. [2011] used OMI data (with IWC > 40 g/kmˆ2) and found very little latitude dependence

in the harmonic fits (although they did not plot change in IWC/brightness vs. latitude, as shown here).

14. p. 14, lines 1-4: Compare this figure to OMI results. The slope between 3-6 h LT is indeed very steep, but it includes many faint PMC and thus potentially larger variations in occurrence frequency.

15. p. 14, lines 7-8: You can also consider the Stevens et al. [2017] discussion regarding definition of occurrence frequency and how it folds into such analysis.

16. p. 14, lines 13-14: Are the differences between these results for A24/A12 and the brightness ratios listed in Table 1 significant? Should the results in Table 2 for 61.5-64.5 N be considered as comparable to the "faint" cloud class in Table 1?

17. p. 14, lines 17-19: You have already discussed the importance of threshold selection (beta_max, IWC) in deriving such local time variations. Can models give some guidance as to whether these variations are more (or less) important in such an analysis (e.g. SOFIE threshold vs. CIPS vs. SBUV)?

18. p. 15, lines 7-8: Recent intervals of 3-4 years in Figure 10(c) with locally larger amplitude and more year-to-year variability (e.g. 1993-1997, 2007-2010) are mostly correlated with solar minimum. Could the internal mechanism for model variations be tied to the level of solar activity?

19. p. 17, lines 13-14: Please add a note that increasing the IWC threshold to satellite measurement levels does change this amplitude significantly. Are different mechanisms (e.g. proportional to number of particles vs. proportional to particle size) more important for either the "no threshold" vs. "satellite threshold" analysis?

20. p. 17, lines 22-23: I don't consider a 4 hour shift "remarkable" here, particularly when the overall variation is a superposition of three harmonic terms.

---

## Referee Comment (RC2) · Anonymous Referee #2 · 12 Nov 2017

General Comments:

This manuscript reports results from the Mesospheric Ice Microphysics And tranSport (MIMAS) model using hourly output prescribed by the Leibniz Institute Middle Atmosphere (LIMA) model in order to draw a variety of conclusions on the variation of Polar Mesospheric Clouds (PMC) over the diurnal cycle. The authors compare their results to a suite of ground-based and satellite PMC datasets and extend their study to include all relevant PMC latitudes and cloud classifications. The authors furthermore draw conclusions about long-term trends in the amplitude of the migrating diurnal and semi-diurnal tidal components of PMC ice water content (IWC). The scope of the study is ambitious and if the results are robust, would significantly advance the state of knowledge on the spatial and temporal variation of some of the most important diagnostic

PMC properties.

However, the reviewer is skeptical that MIMAS is properly characterizing the reported PMC variations. Although the model shows agreement with many of the datasets included in the study, the reviewer is suspicious that in many cases the agreement is fortuitous and does not validate the model ice properties or the model inputs. This is because the authors demonstrate a curious disregard of a variety of relevant observational and modeling studies that show quite different results in both the ice properties and the model inputs. The reviewer lists the concerns below.

Specific Comments:

1. The LIMA inputs largely control the variation of cloud properties over the diurnal cycle. Therefore, Section 5.2 ("Atmospheric background conditions") should be moved to the beginning of Section 2 since everything else flows from those results. Figure 6 (left) is especially important to the rest of the study and shows that the variation of temperature over the diurnal cycle is about +/- 1 K at 83 km at 69 N. The amplitude of this variation is in direct contrast to many other studies showing a much larger observed variation of +/- 3-4 K [Singer et al., 2003; Singer et al., 2005; Stevens et al., 2010; McCormack et al., 2014; Stevens et al., 2017]. The authors need to clarify why they believe their results are more reliable than all of these previous studies. If they cannot, then they need to show how their PMC results respond to this larger amplitude of the thermal tide at PMC altitudes.

2. To further clarify comment #1 and for more direct comparison with previous studies, the reviewer requests an additional table (immediately prior to Table 1) showing the tidal variations at the most relevant altitude that enables the PMC variations. The reviewer suggests in rows "All clouds", "faint", "long-term" and "strong" and in columns "T24 (K)", "T12 (K)", "H2O24 (ppmv)" and "H2O12(ppmv)".

3. The authors need to provide additional details on the vertical distribution of condensation nuclei (CN) used in their simulation. There is reference to a Hunten distribution

on page 3, line 4. If they refer to Hunten et al. [1980] they need to cite this work and they also need to evaluate the reliability of their results against more contemporary studies that include global-scale transport, that have much smaller CN densities [Bardeen et al., 2008; Megner et al., 2008; Rapp and Thomas, 2006].

4. On the top of p. 14 (line 1) the authors state that "the amplitude of the local time dependence increases in absolute IWC values towards the pole". Figure 9 is shown in support of this statement. The reviewer does not understand this result and would like an explanation. Are the authors saying that the magnitude of the thermal tide increases toward the pole? If so, that is in direct contrast to previous modeling and observational studies [Chang et al., 2008; Stevens et al., 2017]. If there is some other reason, then they need to state it explicitly.

5. It would be very useful to see a comparison of IWC from CIPS against the results in Figure 8. To the author's knowledge such has a model-data comparison has not yet been done. The authors should also know that Bailey et al. [2015] directly compared CIPS and SOFIE IWC and found CIPS was a factor of 2-3 too low when measuring at the same local time as SOFIE. This is also relevant to their comparison in Figure 3. The values near 80 N look comparable to the results of Stevens et al. (2017) but a large diurnal variation is inferred by the authors and this needs to be discussed in the text.

6. In Section 7 and Figure 10 the authors report a long-term trend in the amplitudes of the diurnal and semi-diurnal tide. To the reviewer's knowledge this has not been shown before. The reviewer is therefore frustrated that the authors reserve their explanation of this for a future study. If they cannot explain what causes this long-term trend, then they need to withdraw this conclusion from the manuscript until they know the cause.

Technical Corrections:

1. General comment. In all figure captions and table captions for IWC, please explicitly indicate whether values of "IWC=0" are included in the results to avoid any confusion.

[Figure]

Some in the field do not weight their IWC with PMC occurrence frequency and others do so it is important to be clear wherever possible.

2. Abstract, p. 1, line 3. Do the authors mean "...good agreement between model and lidar observations at 69 N"? Please be explicit.

3. Abstract, p. 1, line 5. "...from satellite observations" should be clarified. Please state which satellite observations. Also, the AIM satellite is in a sun synchronous orbit so both CIPS and SOFIE observations are locked in local time. Therefore, these observations are not easily tested against results from a model study on local time dependence. That does not mean that the AIM observations should not be used, but the authors need to better clarify how they are used.

4. Abstract, p. 1, line 7. The maximum to minimum ratio is strongly dependent on the threshold used and this need to be clarified here or the statement should be removed.

5. Abstract, p. 1, line 7-8. This conclusion will depend strongly on how the condensation nuclei are prescribed (see specific comment #3 and Rapp and Thomas (2006, Table 1)). If the conclusion is too uncertain given the model inputs then it should be removed.

6. Abstract, p.1, line 8-9. The reviewer is particularly skeptical of the conclusion about the absolute tidal variation increasing to the pole. Please see specific comment #4 and re-evaluate.

7. Abstract, p. 1, lines 9-12. Please see specific comment #6 and re-evaluate.

8. Abstract, lines 12-13. Please see specific comment #1 and re-evaluate. Also, to avoid confusion the authors need to state a temperature amplitude (i.e. +/- X K or +/- X ppmv) and the dominant tidal component.

9. p. 2, line 15. "Opposite to satellites" should be "In contrast to satellite measurements".

10. P. 2, line 32. "with same" should be "with the same".

11. P. 3, line 8. "In case..." should be "In the case..."

12. Figure 1 caption (and throughout manuscript). In order to clearly distinguish what is observed and what is modeled, the reviewer requests that the authors not use the word "data" when reporting their model results. In the middle of the Figure 1 caption therefore "model data" should be "model results" and at the bottom of the Figure 1 caption, "MIMAS data" should be "MIMAS results".

13. P. 8, line 12. In order to avoid all confusion, the authors should state here whether PMC frequency (or IWC=0 values) is included in the IWC results presented. This is clarified later but should be stated here.

14. P. 9, lines 14-15. The reviewer understands what the authors are trying to say, but this could be confusing. After all, if the PMC threshold is raised high enough then there will be no detections at the mimimum so that the maximum/minimum is infinity. Perhaps it would be more clear instead to say "Hence, the strength of the local time variations is sensitive to the PMC occurrence frequency".

15. P. 10, Figure 4. The reviewer is a little skeptical that A24/A8 can be determined to 3 significant figures. Could the authors please expand on their decision to include 3 components? For example, what does the solution look like with only a diurnal and semi-diurnal fit?

16. P. 14, Table 2. It appears from the discussion in the text that no threshold was applied to these numbers. If so, please say so explicitly in the table caption. Also, the numbers for A24/A12 seem quite a bit different from those reported by Stevens et al. (2017) for the same time period. Since the approach to simulating the ice particle formation is quite different between the two studies, it would be illustrative to show A24/A12 for temperature and A24/A12 for H2O, perhaps in a separate table, analogous to the request in specific comment 2.

[Figure]

17. Please re-evaluate and revise the conclusions given the specific and technical comments listed above. Thank you.

References

Bailey, S.M. et al. (2015), Comparing nadir and limb observations of polar mesospheric clouds: The effect of the assumed particle size distribution, J. Atm. Sol.-Terr. Phys., 127, 51-65.

Bardeen, C.G. et al. (2008), Numerical simulations of the three-dimensional distribution of meteoric dust in the mesosphere and upper stratosphere, J. Geophys. Res., 113, D17202, doi:10.1029/2007JD009515.

Chang, L. et al. (2008), Structure of the migrating diurnal tide in the Whole Atmosphere Community Climate Model (WACCM), Adv. Space Res., 41, 1398-1407.

McCormack, J.P. et al. (2014), Intraseasonal and interannual variability of the quasi 2 day wave in the Northern Hemisphere summer mesosphere, J. Geophys. Res. Atmos., 119, 2928-2946, doi: 10.1002/2013JD020199.

Megner, L. et al. (2008), Global and temporal distribution of meteoric smoke: A two-dimensional study, J. Geophys. Res., 113, D03202, doi:10.1029/2007JD009054.

Rapp, M. and G.E. Thomas (2006), Modeling the microphysics of mesospheric ice particles: Assessment of current capabilities and basic sensitivities, J. Atmos. Sol.-Terr. Phys., 68, 715-744.

Singer, W. et al. (2003), Temperature and wind tides around the summer mesopause at middle and arctic latitudes, Adv. Space Res., 31, 2055-2060.

Singer, W. et al. (2005), Tides near the Arctic summer mesopause during the MaCWAVE/MIDAS summer program, Geophys. Res. Lett., 32, L07S90, doi:10.1029/2004GL021607.

Stevens, M.H. et al. (2010), Tidally induced variations of polar mesospheric cloud

altitudes and ice water content using a data assimilation system, J. Geophys. Res., 115, D18209, doi:10.1029/2009JD013225.

Stevens, M.H. et al. (2017), Periodicities of polar mesospheric clouds inferred from a meteorological analysis and forecast system, J. Geophys. Atmos., 122, 4508-4527, doi:10.1002/2016JD025349.

---

## Author Comment (AC1) · 13 Apr 2018

*(Author responses are in italics. Line numbers refer to the revision without tracked changes. In the tracked version deleted sequences are marked red. New text is marked in blue.)*

***General comment to reviewer I***

*We want to thank the two reviewers for the detailed reviews with many useful ideas and suggestions which, we think, have significantly increased the quality of the manuscript.*

*We have rewritten a substantial portion of the manuscript. In particular, we have added three new tables.*

*Table 1: Local time variations of background temperature,*
*Table 2: Local time variations of background water vapor,*
*Table 5: Local time variations of ice water content.*

*We shifted section 5.2 (old: Atmospheric background conditions) to a new section 2.2 (Mean state and local time variations of atmospheric background temperature and water vapor). The new section 2.2 discusses in detail new Tables 1 and 2.*

*We have rewritten section 6 (Latitudinal variations of local time dependence for ice water content) where we now discuss in detail the local time variations of IWC in terms of different thresholds and different latitudes. This includes a new discussion of SBUV thresholds presented in the new Table 5.*

*The abstract and conclusion sections have been adapted. Also, we have included several new references.*

*Finally, we decided to remove the old section 7 (Long-term variations 1997 - 2013) which contained a short presentation of possible trends in tidal IWC amplitudes. The reasons for this withdrawal are:*

*1) This section was rather short, included only one figure, and showed simply a trend behavior of one special IWC parameter, i.e. tidal amplitude, for one latitude and one threshold. The section lacked any discussion and physical interpretation regarding possible sources and causes of such trends.*

*2) We investigated in more detail the subject of trends in local time variations. It turned out that this is a complex topic which certainly needs further investigations. Several parameters, like latitude and thresholds, play a role which needs to nailed down regarding the impact on local time variations of different ice parameters. Furthermore the effects of possible tidal trends in temperature and water vapor have to be taken into account. Having all this in mind, we decided to cover these topics in near future in a separate paper, which appears to be a better and more systematic way compared to the previous manuscript version.*

**Anonymous Referee #1**

SUMMARY
This paper presents an analysis of local time variations in polar mesospheric cloud (PMC) properties using a 3-D atmospheric model (MIMAS). The results are compared to local time variations derived from lidar data at a single location (ALOMAR in Norway), as well as zonal average results from the SOFIE and CIPS instruments on the AIM satellite. MIMAS also calculates many parameters describing the background atmosphere [e.g. temperature, water vapor, ice particle radius] that are examined for their contributions to local time variations.

GENERAL COMMENT: For better or worse, we may never get a satellite measurement of PMCs with simultaneous SOFIE-level sensitivity and comprehensive global coverage. So if these model results are to be validated against satellite data, I think that presenting curves based on some of those higher thresholds would be quite valuable. The authors might wish to primarily use qualitative statements in the main paper, and provide extra figures in an appendix or on-line supplement (since this paper is a "model study"). But since there is the possibility of non-linear behavior in going from no threshold in IWC to a SBUV-type threshold (for example), I think that providing such information somewhere would help the acceptance of the large variations shown in some aspects of this analysis.

*The reviewer addresses an important point about SBUV-type thresholds. The SBUV instrument is typically measuring IWC with a threshold 40 g/km^2. SBUV has observed PMC since 1979. Long term variations in IWC derived from SBUV measurements have been presented by Hervig and Stevens [2014] and DeLand and Thomas [2015].*
*Most important is the local time correction of SBUV data in order to investigate long-term changes in PMC. We decided to add Table 5 and address this point*
*in detail in section 6 (Latitudinal variations of local time dependence for ice water content).*

See new text from page 17, line 23 to page 20, line 10.

This paper is well-written. Some suggestions and comments related to specific items are provided below.

SPECIFIC COMMENTS

1. p. 1, lines 23-24: So the relative strength of these components (where both are present) is actually a guide to lower atmosphere structure? This is relevant to comment #10.

*Yes, according to classical theory published in the textbook by Lindzen and Chapman (1970) the diurnal tide is mainly excited by solar absorption of tropospheric water vapor whereas the semidiurnal tide is mainly excited by solar absorption of stratospheric ozone.*

*We mention the report by Lindzen and Chapman (1970). Please, have a look in the textbook at p. 139 ...show that ozone is considerably more important than water vapor in exciting semidiurnal oscillations. This is because ozone excitation occurs over a greater depth than water vapor excitation, and at higher altitudes...*

*and p.153, ... Thus we are not surprised that the contributions to the modes with negative h's from water vapor (near the ground) are larger than the contributions from ozone (far above the ground). However, the contributions from water vapor absorption to the modes with*

*positive h's are also larger. This is due to the short vertical wavelengths associated with these modes. The ozone excitation is distributed over a very considerable depth of the atmosphere (ca. 40 kin). Thus, waves excited at one level can destructively interfere with waves excited at another level (see Buffer and Small (1963), and Lindzen (1966b), for a more detailed discussion of this process). For the (1,1) mode (wavelength ~ 28 kin) the region of water vapor excitation is not sufficiently thick (ca. 18 km) for this process to be of great importance. This, however, is no longer true for the (1,3) and subsequent modes…*

2. p. 2, lines 13-14: Please clarify that this limitation is due to local time sampling, not spatial coverage.

*Done: This sentence was considered redundant and was removed.*

3. p. 2, lines 15-18: Please note also that in contrast to the previous statement, the restricted spatial coverage of lidar data presents a limitation in terms of how well results from any single location can be generalized to other locations (both latitude and longitude).

*Done:  We insert  … are geographically restricted but …*

4. p. 5, lines 18-22: This seems like a reasonable choice because the model can probably form clouds more easily. However, the next paragraph (e.g. lines 25-27) seems to give a different result. Since local time variations are a perturbation on existing clouds, they presumably indicate increased sensitivity to the effectiveness of formation mechanisms. This sensitivity should be addressed later.

*We agree that local time variations are a perturbation on existing clouds. We discuss this sensitivity in terms of background conditions of temperature and water vapor, see section 6, page 19, line 15.*

As shown in section 2.2, phase positions of minimum temperature at PMC altitudes move to some extent during early morning hours backwards in time in poleward direction. Also the phase of the daily water vapor maximum tends to follow this time shift. We conclude that both temperature and water vapor phases cause the general early morning hour structure in IWC and its shift towards higher latitudes.

5. p. 6, lines 12-13: Please connect this concept to the ideas mentioned on the bottom of page 1 regarding how mesospheric thermal variations are being forced.

*We comment: In the introduction we only wanted to give a basic information about the fact that diurnal and semidiurnal tides can be related to different heating by water vapor and ozone which deserve special consideration.*

6. p. 6, lines 23-25: The magnitude of the model variations is significantly larger than the satellite results. Stevens et al. [2017; J. Geophys. Res. Atmos. 122, doi:10.1002/2016JD025349, Section 3.1] discuss the potential differences depending on whether "zero" values are included in averages, but these differences seem large even when that issue is considered.

*Comment: Here we compare model variations with ground-based lidar backscatter data.*

*A comparison of modeled IWC with satellite data is presented in section 4 which shows that MIMAS values are consistent with those reported by AIM-SOFIE and AIM-CIPS. Second, Stevens et al. (2017), see their Fig.6, published modeled IWC results for different latitudes including also one SOFIE point. We added a new Table 5 (section 6), see our response to your general comment. This allows now to compare our modeled IWC with the Stevens results. We see that both model runs describing the local time variation of IWC with a threshold of 40 g/km^2 have similar absolute values and are consistent, page 18, line 11 to page 19, line 8.*

Recently, Stevens et al. (2017) reported about model results of PMC IWC calculations with the NOGAPS-ALPHA model using a 1-d bulk ice model (Hervig et al., 2009b). The authors show that the IWC is largest at highest latitudes and yields a morning peak between 5 and 7 LT and a late afternoon minimum equatorward of 80 N regardless of threshold. Diurnally averaged IWC values (threshold of 40 g/km2) are near 100 g/km2 and consistent with those calculated by MIMAS. NOGAPS-ALPHA results of IWC over a diurnal cycle show at 68 N a ratio between IWC maximum and minimum 5 of about 1.5 for a threshold of 40 (see Figure 6a,b in Stevens et al. (2017)) similar to a ratio of 1.7 from MIMAS calculations. Concurrently, absolute IWC local time variations in NOGAPS-ALPHA increase towards higher latitudes and are threshold dependent. Again, these features are confirmed by MIMAS.

7. p. 6, lines 28-30: These results can be related to the diurnal and semi-diurnal mechanisms discussed on p. 1.

*Again, see our response to your comment 4 and 5.*

8. p. 9, line 5: This variation in IWC is still much larger than the fit to the SBUV data (~15-20% p-p), even given the uncertainty in that result because of the nature of the local time coverage. This makes me question the strength of the statement "compatible to a high degree" on lines 10-11.

*We are a bit confused since there is neither a fit nor a comparison with SBUV data. We discuss two CIPS and one SOFIE data point. Indeed, the satellite values are compatible with model data. Perhaps you think about the factor of 2. We want to answer that local time variations strongly depend on thresholds, see discussion of new Table 5.*

9. p. 9, lines 14-15: See "General Comment" at the beginning of this review. Does a threshold of 40 g/km^2 reduce the local time variation down to the magnitude shown in DeLand and Thomas [2015]?

*Done: Yes, increasing thresholds will decrease (relative) local time variability, see new discussion of section 6. The results from our new Table 5 show a ratio between maximum and minimum of about 1.7 at latitudes 64°-74°N which might be not too far away from a value of 20%-30% reported by DeLand and Thomas [2015], see their Fig.8, 9 showing ratios of descending and ascending points.*

10. p. 10, line 1: The physical arguments presented on p. 1 imply that large ratio values of A24/A12, as listed here, mean that tropospheric forcing of tidal variations is much more important for PMC formation and growth than stratospheric forcing. Is this an appropriate statement?

*Your conclusion is highly speculative. Tropospheric water vapor and its longitudinal variations, tropospheric cloud coverage through release of latent heat will induce variations in the source strength of the tidal excitation of migrating and non-migrating diurnal tidal component. But propagating upwards, any tidal motion (being a sum of different Hough-modes) will experience different thermal background conditions. Also a variable structure of horizontal winds in vertical direction will vary tidal propagation conditions. Finally, one has to consider all kinds of dissipation mechanisms, e.g. turbulence, infrared cooling, wave-wave interaction with gravity waves etc., that will influence amplitude and phase of tides. So we have to state that the complexity and diversity of all these processes make individual and manual analysis impossible.*

11. p. 12, lines 4-5: This result seems surprising given the discussion of high sensitivity to particle radius on p. 5, lines 13-15. Even a few nm matters with an r^6 dependence. Comments?

*Comment: Here we discuss the PMC parameter of ice mass density with an $n*r^3$ dependence. We simply try to analyze which of the two quantities (n versus r) has a larger relative contribution to local time variation in ice mass density.*
*Also note that the remarks on page 8 , line 19, focus on the discussion of a $n*r^6$ dependence in backscatter.*

12. p. 12, lines 15-16: This seems like a significant variation in PMC altitude, considering the small magnitude of quoted long-term variations in z_PMC by Berger and Lubken [2015].

*The local time variation in PMC altitude is about 500 m. The long term trend in PMC altitude is about -150 m per decade, see by Berger and Lubken [2015] their Fig. 3c. Indeed, local time variations of NLC heights are in a comparable range as long-term trends.*

13. p. 13, lines 16-17: What happens with a higher IWC threshold? DeLand et al. [2011] used OMI data (with IWC > 40 g/km^2) and found very little latitude dependence in the harmonic fits (although they did not plot change in IWC/brightness vs. latitude, as shown here).

*Done: We address this issue in the discussion of our new Table 5, see section 6, that shows the local time dependence of IWC > 40 for three different latitude bands. Our model results suggest that relative effects in local time increase towards the pole.*

14. p. 14, lines 1-4: Compare this figure to OMI results. The slope between 3-6 h LT is indeed very steep, but it includes many faint PMC and thus potentially larger variations in occurrence frequency.

*Done, see page 19, line 9.*

On the other hand, DeLand et al. (2011) published local time observations by the Aura OMI (Ozone Monitoring Instrument) satellite instrument which indicates maximum frequency and albedo values at approximately 9-10 h LT at 70 N for the NH 2007 season, with a smaller amplitude and a slight phase shift to 8 h LT at higher latitudes. Hence, model results from MIMAS deviate to some extent from these satellite measurements for 2007. Here we refer to

some year-to-year variations of phases in MIMAS (not shown here) which might explain to some extent these differences.

15. p. 14, lines 7-8: You can also consider the Stevens et al. [2017] discussion regarding definition of occurrence frequency and how it folds into such analysis.

*In section 6 we now discuss all kinds of threshold, with and without frequency weighting, e.g. see discussion of Table 5, section 6.*

16. p. 14, lines 13-14: Are the differences between these results for A24/A12 and the brightness ratios listed in Table 1 significant? Should the results in Table 2 for 61.5-64.5 N be considered as comparable to the "faint" cloud class in Table 1?

*Comparable is the latitude band for 67.-71 from Table 2 (now Table 4) with Table 1 (now Table 3). But have in mind, Table 4 applies for IWC zero counting, i.e. frequency weighting, whereas Table 3 uses brightness threshold intervals. We think, the identification of IWC with faint clouds is not justified.*

17. p. 14, lines 17-19: You have already discussed the importance of threshold selection (beta_max, IWC) in deriving such local time variations. Can models give some guidance as to whether these variations are more (or less) important in such an analysis (e.g. SOFIE threshold vs. CIPS vs. SBUV)?

*According to this point we have rewritten section 6 (Latitudinal variations of local time dependence for ice water content) where we now discuss in detail the local time variations of IWC in terms of different SBUV thresholds, see Table 5 .*

18. p. 15, lines 7-8: Recent intervals of 3-4 years in Figure 10(c) with locally larger amplitude and more year-to-year variability (e.g. 1993-1997, 2007-2010) are mostly correlated with solar minimum. Could the internal mechanism for model variations be tied to the level of solar activity?

*This section has been removed, see our general comments.*

19. p. 17, lines 13-14: Please add a note that increasing the IWC threshold to satellite measurement levels does change this amplitude significantly. Are different mechanisms (e.g. proportional to number of particles vs. proportional to particle size) more important for either the "no threshold" vs. "satellite threshold" analysis?

*No, we don't see different mechanisms. Your question about thresholds has been answered in the conclusions. See page 21, line 13.*
We calculated a climatology of IWC local time variations from a 35-y average from 1979 to 2013 for different thresholds and latitude bands, which might be useful for satellite data analysis in order to perform local time corrections. Local time variations are found to depend on latitude and threshold conditions. For the latitude band 64–74 N and a threshold of IWC > 0 g/km2 IWC maximum and minimum values occur around 3 LT and 19 LT, respectively, with a ratio maximum to minimum of 6.6. For a threshold of IWC>40 g/km2 the local times for maximum and minimum are identical, but the ratio changes to 1.7. A phase shift exists for the IWC local time behavior towards the pole, which is independent of the threshold value.We find the absolute IWC local time variation to generally increase with latitude.

Furthermore, the IWC maximum moves backward in time from 8 LT at mid latitudes to 2 LT at high latitudes.

20. p. 17, lines 22-23: I don't consider a 4 hour shift "remarkable" here, particularly when the overall variation is a superposition of three harmonic terms.

*Conclusions have been rewritten, 'remarkable' is absent.*

---

## Author Comment (AC2) · 13 Apr 2018

*(Author responses are in italics. Line numbers refer to the revision without tracked changes. In the tracked version deleted sequences are marked red. New text is marked in blue.)*

*General comment to reviewer II*

*We want to thank the two reviewers for the detailed reviews with many useful ideas and suggestions which, we think, have significantly increased the quality of the manuscript.*

*We have rewritten a substantial portion of the manuscript. In particular, we have added three new tables.*

*Table 1: Local time variations of background temperature,*
*Table 2: Local time variations of background water vapor,*
*Table 5: Local time variations of ice water content.*

*We shifted section 5.2 (old: Atmospheric background conditions) to a new section 2.2 (Mean state and local time variations of atmospheric background temperature and water vapor). The new section 2.2 discusses in detail new Tables 1 and 2.*

*We have rewritten section 6 (Latitudinal variations of local time dependence for ice water content) where we now discuss in detail the local time variations of IWC in terms of different thresholds and different latitudes. This includes a new discussion of SBUV thresholds presented in the new Table 5.*

*The abstract and conclusion sections have been adapted. Also, we have included several new references.*

*Finally, we decided to remove the old section 7 (Long-term variations 1997 - 2013) which contained a short presentation of possible trends in tidal IWC amplitudes. The reasons for this withdrawal are:*

*1) This section was rather short, included only one figure, and showed simply a trend behavior of one special IWC parameter, i.e. tidal amplitude, for one latitude and one threshold. The section lacked any discussion and physical interpretation regarding possible sources and causes of such trends.*

*2) We investigated in more detail the subject of trends in local time variations. It turned out that this is a complex topic which certainly needs further investigations. Several parameters, like latitude and thresholds, play a role which needs to nailed down regarding the impact on local time variations of different ice parameters. Furthermore the effects of possible tidal trends in temperature and water vapor have to be taken into account. Having all this in mind, we decided to cover these topics in near future in a separate paper, which appears to be a better and more systematic way compared to the previous manuscript version.*

**Anonymous Referee #2**

**General Comments:**
This manuscript reports results from the Mesospheric Ice Microphysics And tranSport (MIMAS) model using hourly output prescribed by the Leibniz Institute Middle Atmosphere (LIMA) model in order to draw a variety of conclusions on the variation of Polar Mesospheric Clouds (PMC) over the diurnal cycle. The authors compare their results to a suite of ground-based and satellite PMC datasets and extend their study to include all relevant PMC latitudes and cloud classifications. The authors furthermore draw conclusions about long-term trends in the amplitude of the migrating diurnal and semi-diurnal tidal components of PMC ice water content (IWC). The scope of the study is ambitious and if the results are robust, would significantly advance the state of knowledge on the spatial and temporal variation of some of the most important diagnostic PMC properties.
However, the reviewer is skeptical that MIMAS is properly characterizing the reported PMC variations. Although the model shows agreement with many of the datasets included in the study, the reviewer is suspicious that in many cases the agreement is fortuitous and does not validate the model ice properties or the model inputs. This is because the authors demonstrate a curious disregard of a variety of relevant observational and modeling studies that show quite different results in both the ice properties and the model inputs. The reviewer lists the concerns below.

**Specific Comments:**

1. The LIMA inputs largely control the variation of cloud properties over the diurnal cycle. Therefore, Section 5.2 ("Atmospheric background conditions") should be moved to the beginning of Section 2 since everything else flows from those results.

*Done, we shifted this section to section 2, see our general comments.*

Figure 6 (left) is especially important to the rest of the study and shows that the variation of temperature over the diurnal cycle is about +/- 1 K at 83 km at 69 N. The amplitude of this variation is in direct contrast to many other studies showing a much larger observed variation of +/- 3-4 K [Singer et al., 2003; Singer et al., 2005; Stevens et al., 2010; McCormack et al., 2014; Stevens et al., 2017]. The authors need to clarify why they believe their results are more reliable than all of these previous studies. If they cannot, then they need to show how their PMC results respond to this larger amplitude of the thermal tide at PMC altitudes.

*Done, we also included a discussion of MIMAS inputs. We discussed local time variations of temperature and water vapor. We reference* [Singer et al., 2003; Singer et al., 2005; Stevens et al., 2010; Stevens et al., 2017] *and we compare tidal amplitudes, see discussion of Tables 1 and 2.*

2. To further clarify comment #1 and for more direct comparison with previous studies, the reviewer requests an additional table (immediately prior to Table 1) showing the tidal variations at the most relevant altitude that enables the PMC variations. The reviewer suggests in rows "All clouds", "faint", "long-term" and "strong" and in columns "T24 (K)", "T12 (K)", "H2O24 (ppmv)" and "H2O12(ppmv)".

*Done, these are the new Tables 1,2 and 5.*

3. The authors need to provide additional details on the vertical distribution of condensation nuclei (CN) used in their simulation. There is reference to a Hunten distribution on page 3, line 4. If they refer to Hunten et al. [1980] they need to cite this work and they also need to evaluate the reliability of their results against more contemporary studies that include global-scale transport, that have much smaller CN densities [Bardeen et al., 2008; Megner et al., 2008; Rapp and Thomas, 2006].

*In section 2.1 (MIMAS model description), page 3, line 12-13, we cite several references [von Zahn and Berger, 2003; Kiliani, 2014; Berger and Lübken, 2015]. We think that dust initialization is not of overriding importance. In MIMAS dust particles are initialized at mesopause heights and are quickly distributed over height regions typically at 84 – 93 km due. Besides 3-d transport, dust particle are affected by particle diffusion which provides an efficient vertical mixing. Secondly, as soon as dust particles are transported outside of an predefined spatial ice model domain (z<83 km, z>93 km, in latitude direction southward of 50N) these particles are randomly relocated into the ice domain near mesopause heights. This process ensures that during a complete ice season dust particles are always available. Of course there is an interaction between ice particles and dust particles. The more ice particles are formed the less dust particle are in total available since the total sum of ice and dust particles is limited to 40 million particles.*

4. On the top of p. 14 (line 1) the authors state that "the amplitude of the local time dependence increases in absolute IWC values towards the pole". Figure 9 is shown in support of this statement. The reviewer does not understand this result and would like an explanation. Are the authors saying that the magnitude of the thermal tide increases toward the pole? If so, that is in direct contrast to previous modeling and observational studies [Chang et al., 2008; Stevens et al., 2017]. If there is some other reason, then they need to state it explicitly.

*No, the thermal tide is decreasing towards the pole, but the water vapor tide increases in poleward direction. For more details see discussion of Tables 1 and 2 (section 2.2) and discussion of Table 5 (section 6).*

5. It would be very useful to see a comparison of IWC from CIPS against the results in Figure 8. To the author's knowledge such has a model-data comparison has not yet been done. The authors should also know that Bailey et al. [2015] directly compared CIPS and SOFIE IWC and found CIPS was a factor of 2-3 too low when measuring at the same local time as SOFIE. This is also relevant to their comparison in Figure 3. The values near 80 N look comparable to the results of Stevens et al. (2017) but a large diurnal variation is inferred by the authors and this needs to be discussed in the text.

*We agree with the reviewer. We also note that Bailey et al. [2015] find remarkable differences between SOFIE and CIPS IWC. We made some new analysis of CIPS data and find that the local dependence in CIPS data, both for ascending and descending branches, depends on latitude and varies from year to year too, see our supplementary plot (cips4.pdf). Hence, a precise comparison of CIPS data versus MIMAS results requires a comprehensive analysis including multiple plots. We think that up to now such a task is beyond the content of our actual paper.*
*But we will perform such an analysis in details in future.*
*We also included a discussion of Stevens 2017 results, see page 18, line 11 to page 19, line 8.*

Recently, Stevens et al. (2017) reported about model results of PMC IWC calculations with the NOGAPS-ALPHA model using a 1-d bulk ice model (Hervig et al., 2009b). The authors show that the IWC is largest at highest latitudes and yields a morning peak between 5 and 7 LT and a late afternoon minimum equatorward of 80 N regardless of threshold. Diurnally averaged IWC values (threshold of 40 g/km2) are near 100 g/km2 and consistent with those calculated by MIMAS. NOGAPS-ALPHA results of IWC over a diurnal cycle show at 68 N a ratio between IWC maximum and minimum 5 of about 1.5 for a threshold of 40 (see Figure 6a,b in Stevens et al. (2017)) similar to a ratio of 1.7 from MIMAS calculations. Concurrently, absolute IWC local time variations in NOGAPS-ALPHA increase towards higher latitudes and are threshold dependent. Again, these features are confirmed by MIMAS.

6. In Section 7 and Figure 10 the authors report a long-term trend in the amplitudes of the diurnal and semi-diurnal tide. To the reviewer's knowledge this has not been shown before. The reviewer is therefore frustrated that the authors reserve their explanation of this for a future study. If they cannot explain what causes this long-term trend, then they need to withdraw this conclusion from the manuscript until they know the cause.

*This is perfectly true. We removed this section, see our general comments.*

**Technical Corrections:**

1. General comment. In all figure captions and table captions for IWC, please explicitly indicate whether values of "IWC=0" are included in the results to avoid any confusion. Some in the field do not weight their IWC with PMC occurrence frequency and others do so it is important to be clear wherever possible.

*Done. We included in all figure captions and table captions the IWC threshold and the information about zero counting (frequeny weighting).*

*In the following comments 2-8 all refer to the abstract. We have completely rewritten the abstract and the conclusions.*

2. Abstract, p. 1, line 3. Do the authors mean ": : :good agreement between model and lidar observations at 69 N"? Please be explicit.

3. Abstract, p. 1, line 5. ": : :from satellite observations" should be clarified. Please state which satellite observations. Also, the AIM satellite is in a sun synchronous orbit so both CIPS and SOFIE observations are locked in local time. Therefore, these observations are not easily tested against results from a model study on local time dependence.
That does not mean that the AIM observations should not be used, but the authors need to better clarify how they are used.
4. Abstract, p. 1, line 7. The maximum to minimum ratio is strongly dependent on the threshold used and this need to be clarified here or the statement should be removed.

5. Abstract, p. 1, line 7-8. This conclusion will depend strongly on how the condensation nuclei are prescribed (see specific comment #3 and Rapp and Thomas (2006, Table 1)). If the conclusion is too uncertain given the model inputs then it should be removed.

6. Abstract, p.1, line 8-9. The reviewer is particularly skeptical of the conclusion about the absolute tidal variation increasing to the pole. Please see specific comment #4 and re-evaluate.

7. Abstract, p. 1, lines 9-12. Please see specific comment #6 and re-evaluate.

8. Abstract, lines 12-13. Please see specific comment #1 and re-evaluate. Also, to avoid confusion the authors need to state a temperature amplitude (i.e. +/- X K or +/- X ppmv) and the dominant tidal component.

9. p. 2, line 15. "Opposite to satellites" should be "In contrast to satellite measurements".

*Done.*

10. P. 2, line 32. "with same" should be "with the same".

*Done.*

11. P. 3, line 8. "In case: : :" should be "In the case: : :"

*Done.*

12. Figure 1 caption (and throughout manuscript). In order to clearly distinguish what is observed and what is modeled, the reviewer requests that the authors not use the word "data" when reporting their model results. In the middle of the Figure 1 caption therefore "model data" should be "model results" and at the bottom of the Figure 1 caption, "MIMAS data" should be "MIMAS results".

*Done.*

13. P. 8, line 12. In order to avoid all confusion, the authors should state here whether PMC frequency (or IWC=0 values) is included in the IWC results presented. This is clarified later but should be stated here.

*At all discussion points, now, we always state which counting and threshold method has been applied.*

14. P. 9, lines 14-15. The reviewer understands what the authors are trying to say, but this could be confusing. After all, if the PMC threshold is raised high enough then there will be no detections at the minimum so that the maximum/minimum is infinity. Perhaps it would be more clear instead to say "Hence, the strength of the local time variations is sensitive to the PMC occurrence frequency".

*We think that the ratio between maximum and minimum is a reasonable parameter. Of course, this ratio should be well defined.*

15. P. 10, Figure 4. The reviewer is a little skeptical that A24/A8 can be determined to 3 significant figures. Could the authors please expand on their decision to include 3

components? For example, what does the solution look like with only a diurnal and semi-diurnal fit?

*The reviewer is right. The fit curve would be almost identical using only a 24 h and a 12 h fit. Note that all new Tables 1,2, and 5 contain only diurnal and semidiurnal information indicating that the terdiurnal mode is of minor importance. E.g. we included such a statement at page 15, line 1.*

The fit is dominated by the diurnal and semidiurnal mode, the terdiurnal mode is of minor importance.

16. P. 14, Table 2. It appears from the discussion in the text that no threshold was applied to these numbers. If so, please say so explicitly in the table caption. Also, the numbers for A24/A12 seem quite a bit different from those reported by Stevens et al. (2017) for the same time period. Since the approach to simulating the ice particle formation is quite different between the two studies, it would be illustrative to show A24/A12 for temperature and A24/A12 for H2O, perhaps in a separate table, analogous to the request in specific comment 2.

*The reviewer is right. No threshold was applied, and zero counting (frequency weighting) has been used. We added a new Table 5 (section 6), see our response to your general comment. This allows now to compare our modeled IWC with the Stevens results. We see that both model runs describing the local time variation of IWC with a threshold of 40 g/km^2 have similar absolute values and are consistent, page 18, line 11 to page 19, line 8.*

Recently, Stevens et al. (2017) reported about model results of PMC IWC calculations with the NOGAPS-ALPHA model using a 1-d bulk ice model (Hervig et al., 2009b). The authors show that the IWC is largest at highest latitudes and yields a morning peak between 5 and 7 LT and a late afternoon minimum equatorward of 80 N regardless of threshold. Diurnally averaged IWC values (threshold of 40 g/km2) are near 100 g/km2 and consistent with those calculated by MIMAS. NOGAPS-ALPHA results of IWC over a diurnal cycle show at 68 N a ratio between IWC maximum and minimum 5 of about 1.5 for a threshold of 40 (see Figure 6a,b in Stevens et al. (2017)) similar to a ratio of 1.7 from MIMAS calculations. Concurrently, absolute IWC local time variations in NOGAPS-ALPHA increase towards higher latitudes and are threshold dependent. Again, these features are confirmed by MIMAS.

17. Please re-evaluate and revise the conclusions given the specific and technical comments listed above. Thank you.

*Conclusions have been revised. We also included a multiple of new references which can be identified in the colored track version. We also thank again for this very precise and detailed review. We know that perhaps we have not answered everything within 100 percent. But nevertheless we hope that the reviewer should have now a larger confidence to MIMAS model results.*

References:
Bailey, S.M. et al. (2015), Comparing nadir and limb observations of polar mesospheric clouds: The effect of the assumed particle size distribution, J. Atm. Sol.-Terr. Phys., 127, 51-65.
Bardeen, C.G. et al. (2008), Numerical simulations of the three-dimensional distribution of meteoric dust in the mesosphere and upper stratosphere, J. Geophys. Res., 113,

D17202, doi:10.1029/2007JD009515.

Chang, L. et al. (2008), Structure of the migrating diurnal tide in the Whole Atmosphere Community Climate Model (WACCM), Adv. Space Res., 41, 1398-1407.

McCormack, J.P. et al. (2014), Intraseasonal and interannual variability of the quasi 2 day wave in the Northern Hemisphere summer mesosphere, J. Geophys. Res. Atmos., 119, 2928-2946, doi: 10.1002/2013JD020199.

Megner, L. et al. (2008), Global and temporal distribution of meteoric smoke: A twodimensional

study, J. Geophys. Res., 113, D03202, doi:10.1029/2007JD009054.

Rapp, M. and G.E. Thomas (2006), Modeling the microphysics of mesospheric ice particles: Assessment of current capabilities and basic sensitivities, J. Atmos. Sol.-Terr. Phys., 68, 715-744.

Singer, W. et al. (2003), Temperature and wind tides around the summer mesopause at middle and arctic latitudes, Adv. Space Res., 31, 2055-2060.

Singer, W. et al. (2005), Tides near the Arctic summer mesopause during the MaCWAVE/MIDAS summer program, Geophys. Res. Lett., 32, L07S90, doi:10.1029/2004GL021607.

Stevens, M.H. et al. (2010), Tidally induced variations of polar mesospheric cloud altitudes and ice water content using a data assimilation system, J. Geophys. Res., 115, D18209, doi:10.1029/2009JD013225.

Stevens, M.H. et al. (2017), Periodicities of polar mesospheric clouds inferred from a meteorological analysis and forecast system, J. Geophys. Atmos., 122, 4508-4527, doi:10.1002/2016JD025349.

---

## Author Response (AR2)

Response to the second report of the reviewer II to our paper:

Title:          Local Time Dependence of Polar Mesospheric Clouds: A model study
Author(s):      Francie Schmidt et al.
MS No.:         acp-2017-772
Iteration:      Minor Revision
Special Issue: Sources, propagation, dissipation and impact of gravity waves (ACP/AMT inter-journal SI)

Again, we thank the reviewer and appreciate his/her suggestion to basically accept our paper for publication with some minor changes. We have taken his/her suggestions for improvements into account when preparing the revised version of the manuscript. Here is our point by point response to the reviewer's comments. We have marked the changes in the revised version of the manuscript.

(*Author responses are in italics. In the tracked version deleted sequences are marked* red. *New text is marked in* blue.)
* * *
***Comment to reviewer II:***

*1. Abstract, lines 12 and 13. In order to avoid any ambiguity, instead of reporting "variations of temperature at 69 N are in the range of 6 K" it is more clear and concise to state "variations of temperature at 69 N are +/-3 K". That way there is no confusion about the amplitude of the variations. Similarly, "the water vapor variation is about 7 ppmv" should instead be "the water vapor variation is about +/-3.5 ppmv". This also applies to the conclusions on p. 21, lines 10 and 11. Please also use this more explicit text for the paragraph on p. 6 lines 4-14 and anywhere else in the manuscript were there could be ambiguity. Thank you.*

Our reply: Done.

*2. P. 5, line 21 to p. 6 line 3. These three sentences are a rather significant dismissal of the cited work. Note that Stevens et al. (2017) use a relatively new version of the SOFIE temperatures that removed a warm bias at the mesopause present in earlier versions. In addition, the reviewer sees very little difference between the ~140 K SOFIE temperature reported at 90 km and temperatures reported at 90 km by Singer et al. (2003, their Figures 2 and 3 for 69 N). Furthermore, although the July 2003 data appear to be have lower temperatures in Singer et al. (2005, their Figure 2), the July 2002 data appear to less than 10 K different than reported July 2009 data from SOFIE. If the authors know of a specific reference that indicates the July 2009 SOFIE data reported by Stevens et al. (2017) is biased ~10 K high at 90 km, please provide it. Otherwise please revise the text here to more accurately reflect the published record. Thank you.*

Our reply: We apologize for any misleading wording. We have modified and clarified the sentences, as suggested.
New Text:  Stevens et al. (2017), see their Fig. 1, also published temperatures at 68°N (23.1-23.3 LST) for July 2009 observed by the SOFIE satellite instrument which show similar temperatures compared to rocket measurements. For example, SOFIE temperatures indicate a mesopause at 88km with a mesopause temperature of ∼ 135 K.

We deleted the sentence: We note that such a warm mesopause would dramatically prevent ice nucleation and growth in MIMAS with resulting highly underestimated ice masses.

*3. Figure 5. It appears that IWC values of zero (or less than the indicated threshold) are not included in this figure. If that is correct, then please say so in the caption to avoid any confusion. The authors are now clearer about this elsewhere in the text.*

Our reply:  In the legend of Fig. 5 we have already indicated a threshold value for IWC of 10. This automatically implies that are values smaller than this threshold are disregarded.

*Also, please choose different symbols to more clearly distinguish the ascending and descending node data for CIPS in Figure 5. Finally, please indicate the version of CIPS data used, as they have done for the SOFIE data. Thank you*

Our reply: Done.

[revised manuscript text omitted]